# Thermodynamic stability of ligand-protected metal nanoclusters

Michael G. Taylor[1] & Giannis Mpourmpakis[1]

Despite the great advances in synthesis and structural determination of atomically precise, thiolate-protected metal nanoclusters, our understanding of the driving forces for their colloidal stabilization is very limited. Currently there is a lack of models able to describe the thermodynamic stability of these 'magic-number' colloidal nanoclusters as a function of their atomic-level structural characteristics. Herein, we introduce the thermodynamic stability theory, derived from first principles, which is able to address stability of thiolate-protected metal nanoclusters as a function of the number of metal core atoms and thiolates on the nanocluster shell. Surprisingly, we reveal a fine energy balance between the core cohesive energy and the shell-to-core binding energy that appears to drive nanocluster stabilization. Our theory applies to both charged and neutral systems and captures a large number of experimental observations. Importantly, it opens new avenues for accelerating the discovery of stable, atomically precise, colloidal metal nanoclusters.

[1] Department of Chemical Engineering, University of Pittsburgh, Pittsburgh, Pennsylvania 15261, USA. Correspondence and requests for materials should be addressed to G.M. (email: gmpourmp@pitt.edu).

Metal nanoclusters (NCs) are an exciting class of materials due to their unique properties that differ from both bulk and atomic-scale behaviour. Colloidal NCs stabilized by the presence of thiolate molecules on their surface, in particular, have broad applications that range from biolabeling to targeted drug delivery to catalysis[1–3]. In the Brust–Schiffrin-type syntheses of these colloidal NCs, metal salts (most notably, Au) are reduced in the presence of thiolate ligands to produce NCs of specific sizes depending on the ligands and reaction conditions used[4,5]. The resulting size (and shape) of the NCs, in turn, determines their physicochemical properties[2]. Advances in materials characterization have enabled the crystal structure determination of a series of thermally stable (magic-number) thiolated metal NCs ($M_nSR_m$, where M = metal and SR = thiolate group) consisting of up to a few hundred atoms[5–19]. First-principles-based computational modelling has also been employed to probe structural and electronic properties of these magic-number clusters. In particular, the 'divide-and-protect' theory emerged in an effort to rationalize the observed structural characteristics of Au NC and the 'superatom' theory to explain the magic-number NC stability[11,14,15,17,20,21].

The divide-and-protect theory suggests that Au NCs form from maximizing Au–Au and Au–S interactions that take place in the core and on the surface of the NC, respectively. This leads to NC structures consisting of metallic Au cores with shell structures (also reported in the literature as cages) formed from thiolate–Au bond networks, –SR(–Au–SR)$_n$–, known as 'staple motifs'[15,16,22]. Divide-and-protect (or similar 'core-in-cage') theory has proven very effective in suggesting NC structural characteristics, with every experimentally identified NC having this core-in-shell structure[15,23]. The superatom theory, on the other hand, states that magic-number stability results from the formation of closed-shell electronic orbital structures, similar to noble gases[17]. This theory has been successful in explaining the optical and catalytic properties of several magic-number NCs, but has been shown to have weaknesses as a universal descriptor for the thermodynamic stability of thiolated Au NCs[14,24]. In particular, the $Au_{20}SR_{16}$ and $Au_{36}SR_{24}$ do not fall in the predictions of the superatom theory, but have been successfully experimentally synthesized and isolated under thermodynamic conditions[25,26]. In addition, although this theory was originally derived solely based on Au NCs, it should theoretically apply to all metals that fall on the same column of the periodic table, since it applies electron counting and shell closure rules. Yet, metals that fall in the same periodic table column (for example, Au versus Cu) do not form NCs of the same size (number of metal atoms and ligands) and structure. Moreover, the divide-and-protect theory only suggests structural trends and does not introduce a quantitative descriptor for NC stability, resulting in theoretical predictions of NC structures that are based on general structural criteria. In this context we define structure as composition (Au versus S content) in addition to NC size and shape (morphology). As a result, the reported computationally predicted NCs have deviated from the experimentally synthesized ones as in the case of $Au_{24}(SR)_{20}$ (refs 11,12). Beyond first-principles calculations, simple geometric scaling laws relating the total number of Au atoms ($n$) to the number of ligands ($m$) in NCs have been discovered, though these relations show limitations in predicting NC morphology[27,28]. In addition, the *in silico* structural prediction of stable NCs is currently computationally intractable for NC sizes larger than a couple of hundred atoms (treated with first-principles methods). Taking all these observations together, there is a critical need to develop theoretical models able to describe the stability of colloidal NCs as a function of the specific NC structural characteristics.

Herein, we propose a 'thermodynamic stability' theory based on first-principles density functional theory (DFT) calculations performed on experimentally identified metal NCs. Our developed theory introduces thermodynamic descriptors that are dependent on the detailed structural characteristics of the NCs. Moreover, our theory introduces new pathways for discovering *in silico* atomic-precise metal NC architectures that are thermodynamically stable and synthesizable in the lab.

## Results

**Thermodynamic stability theory**. Figure 1 highlights all DFT-optimized Au nanostructures along with the designation of which atoms are part of the core or shell. We note that the definition of core and shell metal atoms agrees with previous work[25,29–35] with the exception of $Au_{18}SR_{14}$ and $Au_{102}SR_{44}$, where the natural bond orbital charge analysis and S-bonding methods revealed that the core could be more precisely defined by 8 atoms rather than 9 and 77 rather than 79 (see analysis in Supplementary Fig. 2 and Supplementary Note 1)[29]. Based on the core and shell determinations, we calculated the shell-to-core binding energy (BE) and the cohesive energy (CE) of the metal core (see Methods for detailed description). In Fig. 2, we plot the calculated shell-to-core BE versus the CE of the cores for the experimentally determined thiolate-protected Au NCs (points coloured in gold). Interestingly, we reveal a near-perfect match between the shell-to-core BE and the CE of the metal cores. This trend highlights a unique physicochemical feature of the experimentally synthesized $Au_n(SR)_m$ NCs: in order for a thiolate-protected Au NC to be thermodynamically stable, there is a fine balance between the CE of the core and the BE of the shell to the core. Thus, a stability criterion has been elucidated connecting the cores with the shells of the NCs. Interesting enough is the observation that this criterion applies to both neutral (Fig. 1(a)–(i)) and charged (Fig. 2(i)) NCs. In addition, the two structures which would not be identified as stable by the superatom theory, $Au_{20}SR_{16}$ and the $Au_{36}SR_{24}$, are shown as stable here by the thermodynamic stability theory. To test if our model can be extended to other metals, we performed the same analysis for the $[Ag_{25}(SPhMe_2)_{18}]^-$ NC (Fig. 2(ii)) which has been experimentally synthesized[36]. As shown in Fig. 2, again, the CE of the core and the BE of the shell to the core strike a perfect energy balance (see silver point on parity graph). It should be noticed that the Ag NC is negatively charged as in the case of the $[Au_{25}SR_{18}]^-$ NC (Fig. 2(i)), verifying not only that this trend holds for different metals, but also when these metals are charged. As an additional validation test, we created the $[Cu_{25}SR_{18}]^-$ NC (Fig. 2(iii)) based on the crystallographic structure of the corresponding $[Au_{25}SR_{18}]^-$ NC. It is worth mentioning that according to the superatom theory the $[Cu_{25}SR_{18}]^-$ should be a thermodynamically stable nanostructure since Cu and Au are metals with similar electronic shell closure (same period metals). However, the $[Cu_{25}SR_{18}]^-$ has not been experimentally synthesized as of yet, and, we note that that according to our model, the CE of the core dominates the BE of the shell to the core (red point in Fig. 2 deviating from the parity line). While the challenge with synthesizing Cu NCs is largely tied to the persistence of the Cu(I) state[37], our calculation imposes the ideal experimental case where the Cu in $[Cu_{25}SR_{18}]^-$ remains Cu(0). Therefore, we suggest that, at least for this ligand configuration (type and number of ligands), the $[Cu_{25}SR_{18}]^-$ cannot be a magic number NC. We thus believe that the $[Cu_{25}SR_{18}]^-$ serves as a case where the core CE is not balanced with the shell-to-core BE, ruling out this energetic balance as a simple interfacial effect.

To develop a quantitative boundary between synthesizable and non-synthesizable NCs we performed a linear regression on all the experimentally synthesized NCs with 95% confidence and

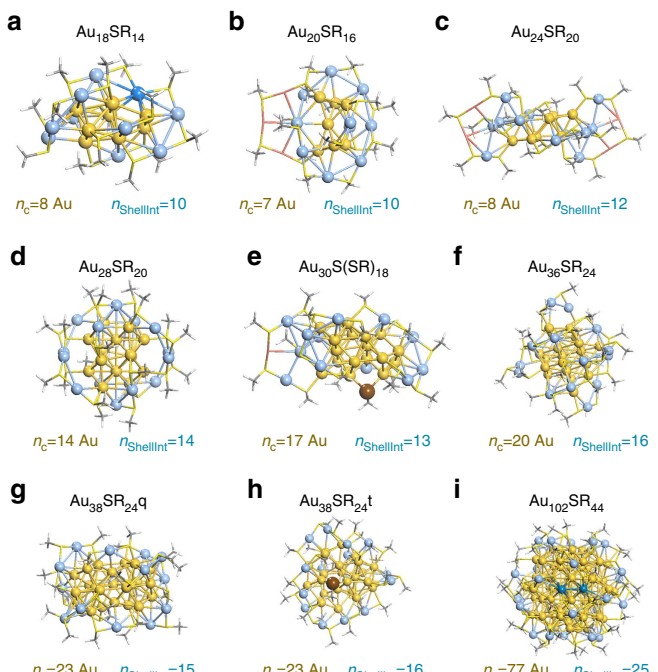

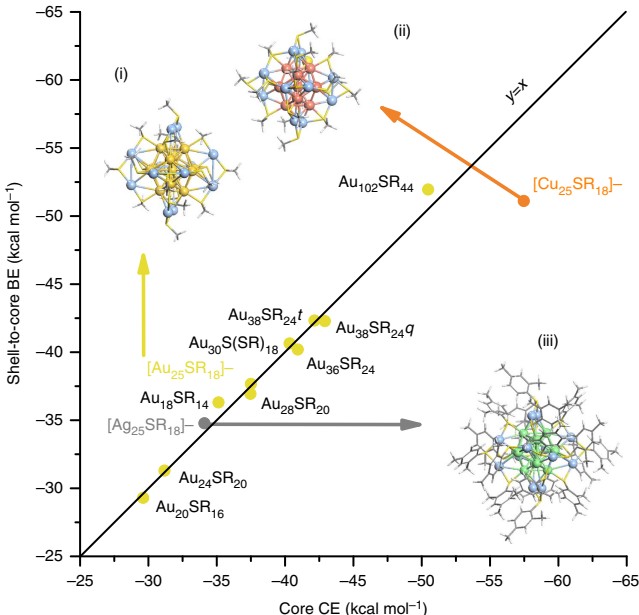

**Figure 1 | Optimized geometries of the experimentally synthesized metal nanoclusters.** (**a**) $Au_{18}SR_{14}$ (ref. 29), (**b**) $Au_{20}SR_{16}$ (ref. 25), (**c**) $Au_{24}SR_{20}$ (ref. 30), (**d**) $Au_{28}SR_{20}$ (ref. 32), (**e**) $Au_{30}S(SR)_{18}$ (ref. 19), (**f**) $Au_{36}SR_{24}$ (ref. 33), (**g**) $Au_{38}SR_{24}q$ (ref. 34), (**h**) $Au_{38}SR_{24}t$ (ref. 65) and (**i**) $Au_{102}SR_{44}$ (ref. 35). $n_c$ represents the number of core metal atoms while $n_{ShellInt}$ represents the number of shell-to-core interactions. Ligands (S-CH$_3$) are shown in stick representation while core and shell atoms, in ball and stick, have been coloured yellow and blue, respectively. In **b,c**, shell Au atoms which do not interact with the core have been coloured red and are shown in stick representation, while in **a,i** shell Au atoms which were previously identified as core are coloured darker blue. In **e,h**, shell sulfur atoms which are not directly bound to a shell Au atom are shown as brown balls.

**Figure 2 | Parity between core cohesive energy and the shell-to-core BE.** The corresponding structures of the $Au_n(SR)_m$ NCs are presented in Fig. 1 except from the optimized structures of (i) $[Au_{25}SR_{18}]^-$ (ref. 31), (ii) $[Cu_{25}SR_{18}]^-$ and (iii) $[Ag_{25}(SPhMe_2)_{18}]^-$ (ref. 36) NCs, which are shown as insets in the graph. For i–iii, $n_c = 13$ metal atoms (Au/Cu/Ag) and $n_{ShellInt} = 12$ as in Fig. 1. The shell metal atoms are shown in blue, whereas, the Cu and Ag core metal atoms are shown in red and green, respectively. Here, all the Au and Ag NCs reported have been experimentally determined. The Cu NC structure is hypothetical, optimized from the Au NC analogous structure (i).

superimposed the prediction bands (Supplementary Fig. 4). To explore the effectiveness of the 95% confidence and prediction bands in distinguishing between non-stable and stable NCs we optimized additional hypothetical NCs. Beyond the hypothetical $[Cu_{25}SR_{18}]^-$ NC, we investigated the $Ag_{18}SR_{14}$, $Cu_{18}SR_{14}$, $Ag_{38}SR_{24}q$ and $Cu_{38}SR_{24}q$ theoretical NCs generated directly from their corresponding Au NC analogues. We found that they exhibit CE and BE values that deviate beyond the 95% prediction band (Supplementary Fig. 5). In addition, we have tested our method on four theoretically predicted Au NCs, the $Au_{18}SR_{14}$, $Au_{20}SR_{16}$, $Au_{24}SR_{20}$ and $Au_{40}SR_{24}$, and showed that two ($Au_{24}SR_{20}$ and $Au_{40}SR_{24}$) out of the four exhibit similar deviation from parity as the theoretical Cu NCs, whereas, the $Au_{18}SR_{14}$ and $Au_{20}SR_{16}$ NCs exhibit the CE and BE energy balance. Therefore, this energetic balance is sensitive to the actual NC structure and the 95% prediction bands can further be used as cutoffs to screen theoretical NCs predicted with current best practices (Supplementary Fig. 5, Supplementary Note 3)[11,38–40].

It should be noticed that when switching the thiolate R group to methyl (to reduce computational cost), attention should be given to the effect that this change introduces to the stability of the shell structure, and in turn, to the shell binding to the core of the NC. Toward understanding ligand impact we highlight that experimentally[41] and theoretically[42], the $[Au_{25}SR_{18}]^-$ NC has been shown to be stable for a wide variety of ligands, and was successfully synthesized even with small, ethyl R groups[43]. Therefore, the exceptional structural stability of $[Au_{25}SR_{18}]^-$

NC seems to be experimentally independent of the ligand type, highlighting the importance of metal structure and $AuS^{-1}$ stoichiometry in determining stable NCs. For NC structures investigated in this work interactions at the interface between their core and shell regions should be to a large degree unaffected by the ligand selection[44] (see Supplementary Fig. 6 and Supplementary Note 4 where $Au_{18}SR_{14}$ and $[Au_{25}SR_{18}]^-$ optimization with full ligands resulted to minor energy shifts and for detailed analysis of the $[Ag_{25}SR_{18}]^-$ case). In addition, metal–metal interactions at the interface are energetically far stronger than the ligand–ligand interactions and capture the core–shell and the relative NC stability. However, enhanced ligand–ligand (R-group) interactions can impact the overall NC stability and associated physicochemical properties as seen in several other recent works[45,46]. For example, in the case of the $[Ag_{25}]^-$ NCs, a π-stacking is observed in the original experimental crystal structures between the phenyl groups present on the shell of the NC. Although the R = methyl group substitution has been shown to have small effect on the RS–Au bond strength[20,47,48], a hydrogen-bond network formed at the NC surface by the groups of the ligands, can potentially induce strain on the shell structure, changing in turn the shell-to-core BE (Supplementary Fig. 6). Interestingly, this observation is in agreement with recent work where conversion from $Au_{38}SR_{24}$ (R = phenylethanethiolate) to $Au_{36}SR_{24}$ (R = 4-tert-butylbenzenethiol) was achieved in solution by swapping the thiolate R groups from phenylethanethiolate to 4-tert-butylbenzenethiol, altering the hydrogen-bond network formed the surface of the NCs[49]. To further prove that this structural thermodynamic stabilization is a general behaviour and originates solely from the energy balance between the core and the shell of the NCs we analysed (Supplementary Note 5) CE

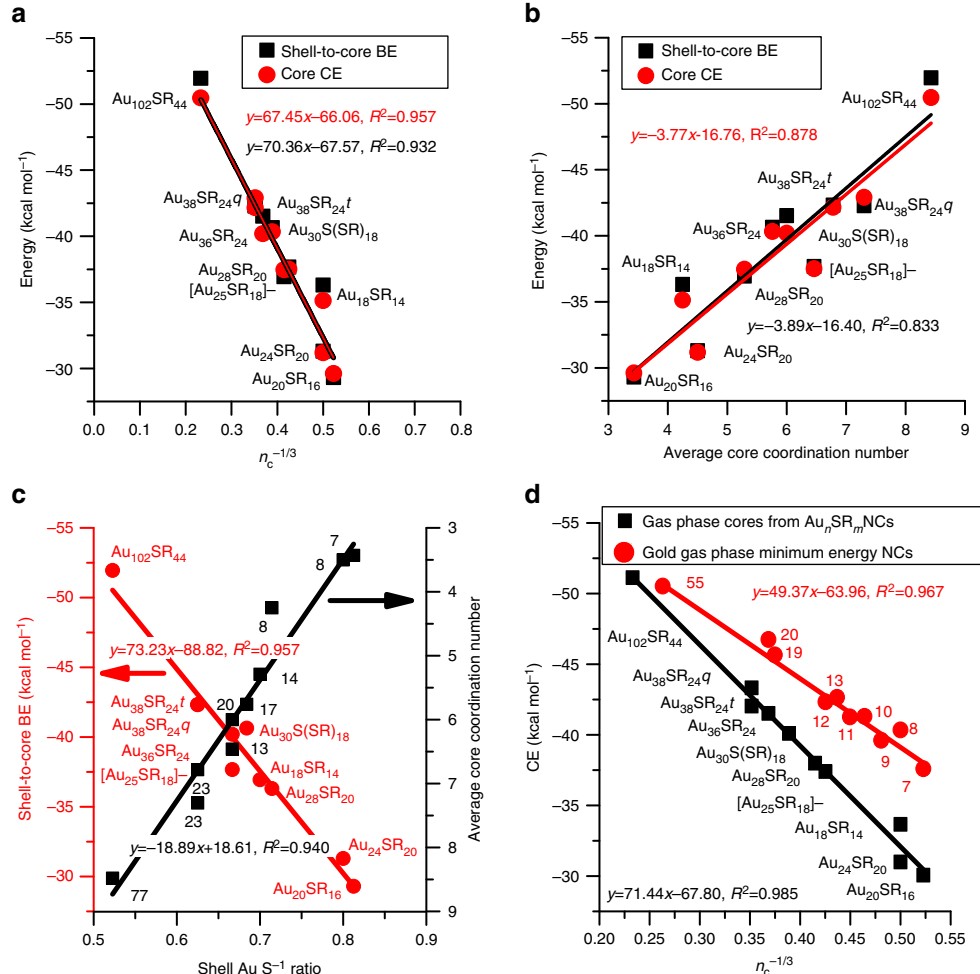

**Figure 3 | Nanocluster stability–morphology relations. (a)** Core CE and shell-to-core BE versus $n_c^{-1/3}$ (number of core metal atoms) for cores of thermodynamically stable magic number Au NCs. **(b)** Core CE and shell-to-core BE versus average coordination numbers (CNs) for cores of Au NCs. **(c)** Shell-to-core BE and average core CN versus the ratio of total Au atoms and S atoms in the shells and **(d)** global minima gas phase Au clusters and cores of Au NCs. From Figs 1 and 2, the Au NC cores contain: $Au_{18}SR_{14} = 8$, $Au_{20}SR_{16} = 7$, $Au_{24}SR_{20} = 8$, $[Au_{25}SR_{18}]^- = 13$, $Au_{28}SR_{14} = 14$, $Au_{30}S(SR)_{14} = 17$, $Au_{36}SR_{24} = 20$, $Au_{38}SR_{24}q$, $t = 23$ and $Au_{102}SR_{44} = 77$ Au atoms.

and BE in the presence of the common[8] dichloromethane (Supplementary Fig. 7(i)) and water (Supplementary Fig. 7(ii)) solvents. The parity between core CE and shell-to-core BE was maintained, with the solvent only weakly affecting the shell-to-core BE. Moreover, we have also tested different DFT methods on a randomly selected system and found that the parity between the core CE and shell-to-core BE were maintained with very slight deviations (Supplementary Table 1). Finally, we present a detailed thermodynamic analysis in Supplementary Note 6 on how this energy balance between the core and the shell relates to the total chemical potential change of the NC ($\Delta\mu(NC) = 0$ at equilibrium), rationalizing the importance of these descriptors and the thermodynamic stability theory.

**Nanocluster size and shape relations.** Because the developed thermodynamic stability theory is based on the morphology-dependent energetic factors of CE of the core and the BE of the shell to the core, we expect these properties to correlate with the structural characteristics of the NCs (that is, size and shape). For example, it is well known that the CE of metals scale linearly with $n_m^{-1/3}$, where $n_m$ is the number of metal atoms in a pure metal cluster. Actually, one can apply first-principles calculations to

derive such linear trends, the limit of which shows the CE of the bulk, when $n_m \rightarrow \infty$, as has been shown in the case of Au (refs 50,51). In Fig. 3a we present such an analysis (core CE versus $n_c^{-1/3}$, where $n_c$ is the number of Au atoms in the core of the NCs) and superimpose the shell-to-core BE results, highlighting the linearity between both energetic factors with $n_c^{-1/3}$ for the thermodynamically stable Au NCs. The reason $n_c^{-1/3}$ trends linearly with CE is attributed to the decrease in the fraction of low-coordinated (surface) sites observed on the NCs as the cluster size increases[51]. Surprisingly, the shell-to-core BE was also found to scale linearly with $n_c^{-1/3}$, with almost identical behaviour (see linear fits) as the CE. The identification of a common structural descriptor for the CE and the shell-to-core BE behaviour on the NCs helps rationalize the observed parity between these two energy contributions in Fig. 2. Since the $n_c^{-1/3}$ shows how the low-coordinated sites scale with NC size (number of metal atoms, $n_c$), then we should expect that the average coordination number (CN) to scale linearly as well with both the CE and the shell-to-core BE. This behaviour is clearly demonstrated in Fig. 3b. The average CN on Au can practically range from 0 (atom) to 12 (bulk). As the average CN of the NC increases, the CE increases (more exothermic values) because the Au atoms tend to form more bonds with their neighbours,

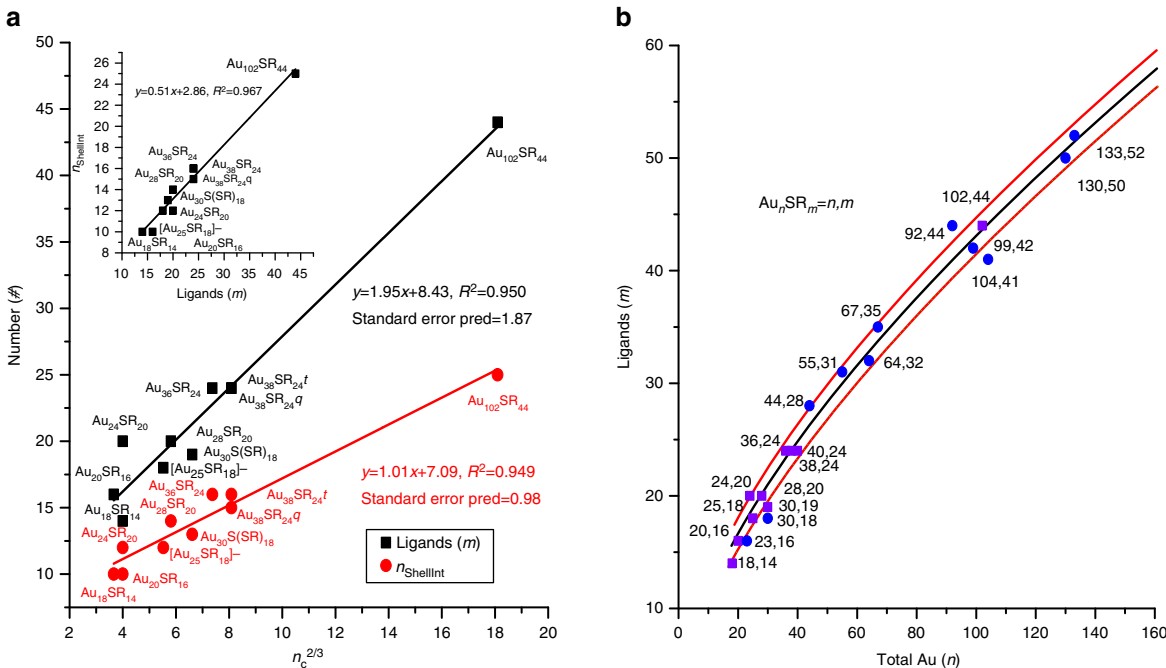

**Figure 4 | Nanocluster stoichiometry relations. (a)** Number of ligands ($m$) and $n_{ShellInt}$ versus $n_c^{2/3}$ for all NCs of Fig. 3. The inset graph shows the $n_{ShellInt}$ versus $m$ behaviour. (**b**) Predicted stoichiometric trend between number of ligands ($m$) and total Au ($n$) atoms of the NCs. The predictions were made using the relations shown in **a**. The black line represents the best fit, whereas, the surrounding red lines the standard error in the prediction. The purple square points represent experimentally stable NCs used in our calculations to develop the model, whereas, the blue circles represent other experimentally stable NCs identified in literature[8].

increasing the overall stability of the NC. On the other hand, as we have recently shown in the area of catalysis, the adsorbates show higher BEs (more exothermic) on sites of the NCs with low CNs[52,53]. However, this is not the case here (see red point data in Fig. 3b). The thiolated-Au shell network binds the core in a way that is counterintuitive to the common belief: as the average CN of the NC increases, the adsorption strength of the shell increases as well. This counterintuitive trend is highlighted by the difference between the predicted and experimental core structures (and resulting deviation from the parity plot) in the $Au_{24}SR_{20}$ NC, where the experimental structure showed a core with lower CN than the predicted structure (Supplementary Fig. 5)[11,30]. In Fig. 3c we plot the shell-to-core BE versus the shell $n_{Au}n_S^{-1}$ ratio (red circles), where $n_{Au}$ is the number of shell Au atoms and $n_S$ is the number of sulfur atoms on the shell (equivalent to $m$ in $Au_nSR_m$). The shell $n_{Au}n_S^{-1}$ ratio demonstrates the cationic character degree of Au on the shell of the NC ($SR^{\delta-}$ interacting with $Au^{\delta+}$) and concentration of bridging thiol groups (SR groups not directly bound to the cores). On the same graph, we plot the average CN of the NC cores versus the $n_{Au}n_S^{-1}$ ratio (black rectangles) on the shell. Notice that both the shell-to-core BE and the core CN scale linearly with the shell $n_{Au}n_S^{-1}$ ratio. It can be observed that the lower the $n_{Au}n_S^{-1}$ ratio, the stronger the shell-to-core BE because of both the increased electrostatic interactions between the core and shell Au atoms (latter are charged more positively) and the decreased amount of bridging thiol groups, which tend to pull shell Au atoms away from the core[47]. On the other hand, the CN versus shell $n_{Au}n_S^{-1}$ ratio linear trend has a negative slope compared to the shell-to-core BE versus $n_{Au}n_S^{-1}$ ratio linear trend. This fact explains why the shell-to-core BE was found to counterintuitively increase as the average core CN increases. This observation was made based on the Au to SR stoichiometry in only the shells of the NCs. Examining the

total Au to SR ratio on the entire NC, we note an overall agreement with the experimental observation of increasing NC diameter resulting from increasing Au to SR ratio in solution[54].

In Fig. 3d, we show the gas phase CE versus $n_c^{-1/3}$ trend for the $Au_nSR_m$ core structures (without the presence of the shells) identified from the crystal structures of the experimentally synthesized NCs (black rectangles) and compare against the CE behaviour of the global minimum energy gas phase Au NC structures of the same size range (red circles). Interestingly, the gas phase CE (equation (3)) is roughly equivalent to the core CE calculated with the presence of the shells in the NCs (see equation (2) and Supplementary Fig. 8). Therefore, the gas phase CEs of the NC cores, accurately represent the stability of the cores in the NC (presence of shell), and can be directly compared with the gas phase global minimum energy structures, in Fig. 3d. The initial structures of the global minimum gas phase clusters were taken from recent literature and were optimized at the same level of theory as the NC cores[55–57]. Figure 3d reveals a difference in the slopes between the minimum energy NCs and the core NC structures. The difference in slopes can be attributed to the morphology imposed on the Au NC cores by the presence of the thiolate shell. Notice that gas phase minimum energy Au clusters preferentially obtain planar structures up to $Au_{13}$, whereas, in the presence of the metal–thiolate shell, they obtain three-dimensional structures[55,58]. We believe that other magic-number thiolated Au NC cores will fall directly on the black line. Overall, Fig. 3 demonstrates for the first time that the stabilization of colloidal NCs in solution is dictated by two thermodynamic descriptors that need to balance: the metal core of the NC tends to grow to increase the CE with NC size (descriptor: CE), while the thiolate–Au network on the shell (acting as adsorbates) obtains a specific composition in staple motifs ($nn_S^{-1}$ ratio),

tuning the shell-to-core BE to match the CE of the core at each NC size.

**Nanocluster stoichiometry relations.** Moving forward, using these relations discovered in Fig. 3, additional stoichiometry rules are needed (that is, $Au_nSR_m$ stoichiometries in addition to core and shell information) to construct a useful methodology for NC prediction. Toward stoichiometry prediction, previous work identified a geometric descriptor based on the surface area to volume ratio of the NCs that relates the number of ligands ($m$) to the total number of Au atoms ($n$) in the NCs with a linear trend of $m$ versus $n^{2/3}$ (refs [27],[28]). For the NCs $n \approx n_c + n_{ShellInt}$ (very small deviations can occur when a shell Au does not bind the core, or a S atom is a direct contact to the core). Given that $n_{ShellInt} \approx n - n_c$ and the $AuS^{-1}$ ratio in the shell dictates a linear trend with shell-to-core BE (Fig. 3c) we would expect $m$ and $n_{ShellInt}$ to be correlated. The inset of Fig. 4a shows that $m$ scales perfectly linearly with $n_{ShellInt}$ ($R^2 = 0.967$). Since $m$ and $n_{ShellInt}$, and $m$ and $n^{2/3}$, are linearly related and since $n_c \approx n - n_{ShellInt}$, a 2/3 exponential relationship (predominates linear functionality) also exists between $n_c$ and $m$ (Fig. 4a). As a result, these observations establish a parametric model for $n$ and $m$ founded on $n_c$. This parametric model, which can now predict the overall NC stoichiometry, is presented in Fig. 4b. We have thus shown (using the relations derived from Fig. 3) that the core morphology largely dictates the overall NC characteristics. Along these lines, our new model captures the previously identified $m$ versus $n^{2/3}$ behaviour and nearly all of the NCs fall within the 95% prediction intervals. Because this model is parametric with $n_c$, however, specific core and shell region information can be immediately derived for NCs of any given $n,m$. For example, given $n_c = 45$, $m \approx 32–34$ and $n_{ShellInt} \approx 19–20$ resulting in the $Au_{64}SR_{32}$, $Au_{65}SR_{34}$, and any other combination between these $n,m$ values to identify NCs (see Fig. 4b for experimentally synthesized $Au_{64}SR_{32}$ NC). From this point, the structure–energy relationships identified in Fig. 3 can be used to feed further structural information to the NC prediction, such as the core CN, as well as to screen candidate structures based on the energy balance criterion between the core CE and the shell-to-core BE (Fig. 2). Thus, the identified relationships aid the prediction of NCs that span sizes larger than the ones currently affordable by high-throughput DFT calculations[18].

## Discussion

In summary, we present a thermodynamic stability theory derived from first-principles calculations, rationalizing the stability of colloidal metal NCs in solution and significantly advancing the previously proposed divide-and-protect and superatom theories[15],[17]. Our theory reveals that for every thermodynamically isolated, experimentally synthesized thiolate-protected NC, there is a perfect energy balance between the adsorption strength of the ligand–shell to the metal–core and the CE of the core. Our theory applies to both neutral and charged NCs, as well as to different metals. In addition, we highlight the impact of the thiolate ligands on the overall stability and size/shape of the NC[5]. Finally, this theory directly relates these thermodynamic stability (energy) contributions to geometrical characteristics of metal cores of the NC, rationalizing NC size and shape effects on NC stability and opening new avenues for *in silico* NC predictions.

## Methods

**Ab initio methodology.** We used the BP-86 (refs [59],[60]) functional combined with the def2-SV(P) basis set[61] accelerated with the resolution of identities approximation[62],[63] as implemented in the Turbomole 6.6 package[64]. Structures were taken directly from previously published work and the R groups of the thiolates were substituted by methyl groups[6],[11],[19],[25],[29–33],[35],[38–40],[65].

conductor-like screening model (COSMO) implicit solvation models were also employed to gauge the effect of dichloromethane ($\varepsilon = 8.93$) solvent on the developed model[66],[67]. The BP-86 functional has been successfully used on thiolated-metal NC systems[48],[68] and the R = methyl group substitution has had little impact on RS–Au bond strength as has been previously applied in computational NC structural determinations[20],[47],[48]. We also note that BP-86 has been successful in capturing stability trends and cohesive energies of very small pure gold clusters[69]. We did not include van der waals corrections in our calculations as they tend to overestimate Au–Au bonding at the interface of Au–thiolate layers[44]. All optimizations were performed without any symmetry constraints.

**Definition of shell-to-core BE.** Two methods were used to identify if Au atoms were 'core' or 'shell', that of natural bond orbital charge analysis and that of measuring S-contacts of Au atoms in the structure, where the shell Au atoms have exactly two bonded sulfur groups (see Supplementary Fig. 1 for example of $Au_{20}$ determination)[70]. These two methods were in perfect agreement over all NCs. With core and shell designations, we isolated the core and shell sections of the NCs and performed single point energy calculations on each section. From the (1) optimized NC structure, (2) separated core and (3) separated shell results, the shell-to-core BE and core CE were calculated. The shell-to-core BE is defined as:

$$Shell - to - Core\ BE = \frac{E_{Full\ Cluster} - E_{Shell} - E_{Core}}{n_{ShellInt}} \quad (1)$$

where $E_x$ = electronic energy of group X and $n_{ShellInt}$ = number of shell contacts interacting with the core (Figs 1 and 2). $n_{ShellInt}$ is largely dictated by the number of shell Au atoms in contact with the surface of the cores ($<4$ Å from the nearest core Au atom) because metal–metal bonds dominate the shell-to-core BE (Supplementary Fig. 3, Supplementary Note 2). Beyond shell Au contacts to the cores, SR groups that are not bound to any shell Au but are bound directly to core Au represent a direct shell-to-core contact and thus are also included in $n_{ShellInt}$ (see Supplementary Note 1 for details surrounding the calculation of $n_{ShellInt}$).

**Definition of core CE.** The core CE is defined as:

$$Core\ CE = \frac{E_{Full\ Cluster} - n_c E_{Metal\ Atom} - E_{Shell}}{n_c + n_{ShellInt}} \quad (2)$$

where $n_c$ = number of metal atoms contained in the core structures (and $E$ is the total electronic energy). For each of the core structures different multiplicities were tested and the lowest-energy spin states were selected for the core CE calculation. For the gas phase minimum energy clusters and NC core structures the CE is defined:

$$CE = \frac{E_{Cluster} - n_c E_{Metal\ Atom}}{n_c} \quad (3)$$

For the core structures, Lennard-Jones radii were used to determine the CNs.

**Data availability.** The datasets generated during and/or analysed during the current study are available from the corresponding author on reasonable request.

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

## Acknowledgements

This material is based upon work supported by the National Science Foundation (CBET-CAREER program) under Grant No. 1652694. M.G.T. acknowledges support by

the National Science Foundation Graduate Research Fellowship under Grant No. 1247842. The authors would like to acknowledge computational support from the Center for Simulation and Modeling (SAM) and the Extreme Science and Engineering Discovery Environment, which is supported by the NSF (ACI-1548562). The authors would also like to thank Professor Rongchao Jin from Carnegie Mellon University, for suggesting experimental structures reported in literature.

## Author contributions

M.G.T. performed all the calculations. G.M. conceived the project and carried out the advising. Both authors aided in the development and writing of this manuscript.

## Additional information

**Competing interests:** The authors declare no competing financial interests.

