## [Peer Review File · Nature Communications]

Reviewers' comments:

Reviewer #1 (Remarks to the Author):

The clusters selected for study in this work are not the most stable sizes experimentally. Au₂₀(SR)₁₆ is reported by only one research group, whereas Au₃₀(SR)₁₈ has been studied by at least 4 research groups, but omitted here.

The authors should consider doing a thorough literature search before carrying out detailed calculations.

"The reason $n^{-1/3}$ trends linearly with CE is attributed to the decrease in the fraction of low-coordinated (surface) sites observed on the NPs as the particle size increases."

The decrease in fraction of surface sites as a function of size (metal atoms) has been reported (a $2/3$ scaling) before thiol monolayer protected clusters. (doi: 10.1021/jp808967p; <http://dx.doi.org/10.1039/C2NR11749E>). A literature search might be useful.

Finally, the term "structural dependent" stability is used in throughout the manuscript heavily, but the structure of the clusters are not taken into account. What the authors mean is more like shape and/or size dependency. This is a major issue.

Overall, this is a good study and the science would be useful.

However, there are major concerns as described above, and it is not clear to me that this manuscript would make a significant impact in the field.

Reviewer #2 (Remarks to the Author):

The authors advocate here a new model that would explain "structure-dependent stability" of ligand-protected metal nanoparticles. The research in this class of materials has become very intense in recent years and thus the paper, if it could live up to expectations, would be very interesting.

However, the title and abstract of this work promise much but the paper does not deliver.

Unfortunately I find several weaknesses and inconsistencies as detailed below. I cannot recommend this for publication in Nature Communication even after major revision, since the authors' model just does not deal with relevant "ingredients" to be really powerful and predictive and thus does not raise such general interest as required for Nature Comm.

1. The dependence of both the core cohesive energy and the shell cohesive energy as $n^{*-1/3}$ is TRIVIAL (fig. 2, the central figure of the paper)! This is simply an interface effect. The number of atoms in the core surface AND the number of atoms at the "inner surface" of the shell dictate the cohesive energy per atom. The number of those atoms go as $n^{*-1/3}$ for an approximate spherical system. Both systems are cut out of the same particle sharing the same interface. That is (simply) why both energies should behave the same way.

2. The paper advocates a "structural model" with predictive power. In fact a simple $n^{*-1/3}$ behavior is found due to the interface effect as discussed above. This model has no predictive power. Why don't the authors use the knowledge and predict here even a few stable compositions? Beyond that comes the structure (is the metal core going to be icosahedral, decahedral, fcc-like, hcp-like, bcc-like... ??) The authors' model says nothing about that, and never can.

3. Besides just re-discovering a well known interface effects in nanoparticles, the authors make a crucial simplification in the calculations. Omitting the real chemical nature of the ligands and resolving

to passivation by SMe only basically neglects all the interactions between the ligands in the ligand layer. The weak organic interactions in the "R-part" of the ligand layer count to several electron volts for tens of ligand and hundreds of carbon atoms and thus provide a significantly increased stabilization effect. The effect is so important that actually there are examples in the literature where the same cluster (same composition) can have different core atomic structure depending on the bulkiness of the ligand. One of the current trends is actually to try to understand the ligand layer structure and the weak interactions in the ligand layer by spectroscopic and computational methods (see, e.g., Nature Comm 7, 10401, 2016).

4. The use of the dielectric model for WATER as a solvent is meaningless. The clusters are computed with the SMe ligand layer, which (if this ligand really could be used in the actual synthesis), would make the cluster material NON-water-soluble.

5. The authors ignore very basic chemical facts when discussing other metals than gold. Some years ago, most of the community thought that only gold can be made stable in nanoparticle form with thiolate or phosphine ligands. Since then several groups have discovered the right chemistry to stabilize a number of silver nanoclusters that have been crystallographically solved to atomic precision (and this referee happens to know that new breakthroughs in that front are going to be published soon). The authors throw a case of the copper cluster $\text{Cu}_{25}(\text{SMe})_{18}(-)$, computed in the hypothetical structure of Au_{25} , as a "negative prediction". Of course, most likely the cluster will not be stable in this structure. It simply might find a better, global minimum, with a suitable (bulky) ligand layer that protects the core from oxidation. Oxidation of copper and the chemistry to protect from it are the challenges (back to point 3).

6. From the title one gets impression that an interface energy analysis would have been made also for silver nanoclusters protected by thiolates. Several crystallographically solved structures are available in the literature ! However this is not done.

7. Comparison of metal core energies to energies of free gas phase metal clusters with the same nuclearity is without a merit. Structures of free gold clusters in gas phase was a hot topic 10-15 years ago. The authors are welcome to review the extensive literature from that time. Structures seen in the metal cores of thiolated gold nanoclusters have little or nothing to do with structures of free gas phase gold clusters of the same nuclearity. This is a fact.

8. A minor point: when talking about the "divide and protect" idea, the authors cite ref. 8. The correct reference is #15 (just look at the title of ref. 15 !).

Reviewer #3 (Remarks to the Author):

This paper reports on computational work used for deriving stability descriptors for ligand-protected gold clusters. The subject is interesting, the approach and the results are original, but the paper itself is not convincing. I have some major concerns at this stage so that I cannot recommend publication.

Major points

a) The key quantities BE and CE are not clearly defined. For what concerns BE, in Eq. (1) it is written that n_{shell} is the number of metal atoms of the shell interacting with the core. This looks quite arbitrary, and in the text it is not explained how these interacting atoms are chosen. Some kind of explanation is given in the comments of the supplementary material, which however do not completely convince about the arbitrariness of the definition. For what concerns CE, there are several unclear points. First of all, is Emetalatoms the total energy of an isolated Au atom? If yes, why

multiply it by $n_{\text{metalatoms}}$? Since core energy is concerned, this should be multiplied by the number of core atoms n_{core} , which should also appear in the denominator of Eq. (2).

b) The derivation of these stability descriptors is not convincingly supported by some line of reasoning. Some kind of thermodynamic reasoning is given in the supplementary material. In my opinion, the derivation of the descriptors should be supported by some physical line of reasoning besides (possible) numerical evidence. The arguments in the supplementary material are only sketched and not convincing. The authors should clearly indicate what are the initial and final states of their thermodynamic reasoning, taking also into account that the core covered by shell atoms in the process, and whether this affects the balance of eq. (5).

Minor points

- The authors do not consider van der Waals corrections. Some checks including them should be made.

- Figure 2c looks rather confusing. Why the values of CN are in decreasing order? If these values are put in increasing order, it is immediately clear that high CN correspond to strong BE.

We would like to thank all three reviewers for their time and very fruitful comments that helped us improve our manuscript. We have significantly revised our manuscript by addressing all the reviewers' comments. Specific responses and actions are listed in this "point-by-point response" file. All changes are highlighted in red in the revised manuscript and supporting information file.

Reviewers' comments:

Reviewer #1 (Remarks to the Author):

The clusters selected for study in this work are not the most stable sizes experimentally. Au₂₀(SR)₁₆ is reported by only one research group, whereas Au₃₀(SR)₁₈ has been studied by at least 4 research groups, but omitted here. The authors should consider doing a thorough literature search before carrying out detailed calculations.

Response: We thank the reviewer for this comment. We are in fact aware of many more nanoclusters (see (1) Jin, R. *Nanoscale* 2015, 15 (7), 1549. and (2) Pichugina, D. A.; Kuz'menko, N. E.; Shestakov, A. F. *Russ. Chem. Rev.* 2015, 84 (11), 1114. for recent extensive reviews) and have selected what we believe to be a representative sample of the stable nanoclusters identified in literature. We highlight that the [Au₂₅(SR)₁₈]⁻, Au₃₈(SR)₂₄, and Au₁₀₂(SR)₄₄ were included in our model, which have been synthesized by different groups. However, we followed the reviewer's advice and have performed additional calculations analyzing the Au₃₀S(SR)₁₈ cluster, which we found to show similar behavior to the other clusters included in our model.

Action:

We have added the new results of our calculations on the Au₃₀S(SR)₁₈ cluster in Figures 1 and 2, along with its relevant reference (1) Crasto, D.; Malola, S.; Brosofsky, G.; Dass, A.; Hakkinen, H. *J. Am. Chem. Soc.* 2014, 136 (13), 5000 (see also *vide-infra* response (b) to reviewer 3).

"The reason $n^{-1/3}$ trends linearly with CE is attributed to the decrease in the fraction of low-coordinated (surface) sites observed on the NPs as the particle size increases." The decrease in fraction of surface sites as a function of size (metal atoms) has been reported (a $2/3$ scaling) before thiol monolayer protected clusters. (doi:10.1021/jp808967p;http://dx.doi.org/10.1039/C2NR11749E). A literature search might be useful.

Response: We thank the reviewer for bringing up this very useful geometric descriptor from the literature. This exponential ($2/3$) scaling for Au_nSR_m that the reviewer has highlighted is derived from taking the surface area to volume ratio, relating the number of ligands (m) to the total number of gold atoms (n) raised to the ($2/3$) power (Area/Volume). While this relation is very useful in rationalizing the overall nanocluster stoichiometry it does not provide information on the: i) energetic (cohesive and binding) trends, ii) core and shell region stoichiometry (shell includes ligands and metal atoms), and iii) morphology of these regions and especially of the core. In the present work we are referring to the core regions of the nanoclusters with the $n^{-1/3}$ relation (commonly used to describe the cohesive energy of particles as their size approaches the bulk – see (1) Haberlen, O. D.; Chung, S. C.; Stener, M.; Rosch, N. *J. Chem. Phys.* 1997, 106 (12), 5189.) to connect energy stability descriptors with geometric descriptors in the core and shell regions of the nanoclusters. However, this purely geometric exponential ($2/3$) relation, kindly introduced to us by the reviewer, has been used to develop an effective prediction

methodology in our revised manuscript. As a result, we have added a relevant discussion and a new Figure (Figure 4) related to thiolate-protected nanocluster prediction based on this (2/3) ratio.

Action:

We have added the following parts in the revised manuscript:

- In the introduction section:

“Beyond first-principles calculations, simple geometric scaling laws relating the total number of Au atoms (n) to the number of ligands (m) in NCs have been discovered, though these relations show limitations in predicting NC morphology.^{25,26}”

“In this context we define structure as composition (Au vs. S content) in addition to NC size and shape (morphology).”

- A new section on the thiolate-protected nanocluster morphology prediction towards the end of the manuscript:

“Moving forward, using these relations discovered in Figure 3, additional stoichiometry rules are needed (i.e. Au_nSR_m stoichiometries in addition to core and shell information) to construct a useful methodology for NC prediction. Toward stoichiometry prediction, previous work identified a geometric descriptor based on the surface area to volume ratio of the NCs that relates the number of ligands (m) to the total number of Au atoms (n) in the NCs with a linear trend of m vs. $n^{2/3}$.^{25,26} For the NCs $n \approx n_c + n_{ShellInt}$ (very small deviations can occur when a shell Au does not bind the core, or a S atom is a direct contact to the core). Given that $n_{ShellInt} \approx n - n_c$ and the Au/S ratio in the shell dictates a linear trend with shell to core BE (Figure 3(c)) we would expect m and $n_{ShellInt}$ to be correlated. The inset of Figure 4(a) shows that m scales perfectly linearly with $n_{ShellInt}$ ($R^2=0.967$). Since m and $n_{ShellInt}$, and m and $n^{2/3}$, are linearly related and since $n_c \approx n - n_{ShellInt}$, a 2/3 exponential relationship (predominates linear functionality) also exists between n_c and m (Figure 4(a)). As a result, these observations establish a parametric model for n and m founded on n_c . This parametric model, which can now predict the overall NC stoichiometry, is presented in Figure 4(b). We have thus shown (using the relations derived from Figure 3) that the core morphology largely dictates the overall NC characteristics. Along these lines, our new model captures the previously identified m vs. $n^{2/3}$ behavior and nearly all of the NCs fall within the 95% prediction intervals. Because this model is parametric with n_c , however, specific core and shell region information can be immediately

Figure 4: (a) Number of Ligands (m) and n_{ShellInt} vs. $n_c^{2/3}$ for all NCs of Figure 3. The inset graph shows the n_{ShellInt} vs. m behavior. (b) Predicted stoichiometric trend between Number of Ligands (m) and Total Au (n) atoms of the NCs. The predictions were made using the relations shown in (a). The black line represents the best fit, whereas, the surrounding red lines the standard error in the prediction. The purple square points represent experimentally stable NCs used in our calculations to develop the model, whereas, the blue circles represent other experimentally stable NCs identified in literature.⁸

derived for NCs of any given n,m. For example, given $n_c=45$, $m \approx 32-34$ and $n_{\text{ShellInt}} \approx 19-20$ resulting in the Au₆₄SR₃₂, Au₆₅SR₃₄, and any other combination between these n,m values to identify NCs (see Figure 4(b) for experimentally synthesized Au₆₄SR₃₂ NC). From this point, the structure-energy relationships identified in Figure 3 can be used to feed further structural information to the NC prediction, such as the Core CN, as well as to screen candidate structures based on the energy balance criterion between the Core CE and the Shell to Core BE (see Figure 2). Thus, the identified relationships aid towards the prediction of NCs that span sizes larger than the ones currently affordable by high-throughput DFT calculations.¹⁸

- To clarify the $n_c^{-1/3}$ scaling relation:

“For example, it is well known that the CE of metals scale linearly with $n_m^{-1/3}$, where n_m is the number of metal atoms in a pure metal cluster.”

Finally, the term "structural dependent" stability is used in throughout the manuscript heavily, but the structure of the clusters are not taken into account. What the authors mean is more like shape and/or size dependency. This is a major issue.

Response: We acknowledge that the definition of structure in the context of metal nanoclusters can be wide, but we have purposely highlighted size/shape dependencies. This was done to enforce the idea that nanocluster morphology and structural characteristics directly influence the nanocluster stability. For instance, a structural descriptor of the nanoclusters that is related to the overall energetic stability is the metal core average coordination number, which is both dependent to the size and shape of the nanoclusters.

Action:

We have added the following clarification sentence to the end of the introduction: “In this context we define structure as composition (Au vs. S content) in addition to NC size and shape (morphology).”

Overall, this is a good study and the science would be useful.

However, there are major concerns as described above, and it is not clear to me that this manuscript would make a significant impact in the field.

Reply: We are thankful for the reviewer’s positive view of our study as a whole and we believe with our current revision the impact of the work has been further communicated.

Reviewer #2 (Remarks to the Author):

The authors advocate here a new model that would explain "structure-dependent stability" of ligand-protected metal nanoparticles. The research in this class of materials has become very intense in recent years and thus the paper, if it could live up to expectations, would be very interesting. However, the title and abstract of this work promise much but the paper does not deliver. Unfortunately, I find several weaknesses and inconsistencies as detailed below. I cannot recommend this for publication in Nature Communication even after major revision, since the authors' model just does not deal with relevant "ingredients" to be really powerful and predictive and thus does not raise such general interest as required for Nature Comm.

1. The dependence of both the core cohesive energy and the shell cohesive energy as $n^{**(-1/3)}$ is TRIVIAL (fig. 2, the central figure of the paper)! This is simply an interface effect. The number of atoms in the core surface AND the number of atoms at the "inner surface" of the shell dictate the cohesive energy per atom. The number of those atoms go as $n^{**(-1/3)}$ for an approximate spherical system. Both systems are cut out of the same particle sharing the same interface. That is (simply) why both energies should behave the same way.

Response: The statement that the balance between the CE and BE is "trivial" because both the core and shells are cut from the same interface is actually shown to be false when analyzing the $[\text{Cu}_{25}\text{SR}_{18}]^-$ compared to the $[\text{Au}_{25}\text{SR}_{18}]^-$ and $[\text{Ag}_{25}\text{SR}_{18}]^-$ nanoclusters (NCs). The experimentally synthesized Au_{25} and Ag_{25} nanoclusters show a parity between the CE and BE, whereas, the equivalent Cu_{25} nanocluster shows significant deviation. As we show in our manuscript this is only the case for thermodynamically stable, experimentally synthesized clusters. To further prove this, we have performed additional calculations on the theoretically predicted $\text{Au}_{18}\text{SR}_{14}$, $\text{Au}_{20}\text{SR}_{16}$, $\text{Au}_{24}\text{SR}_{20}$, and $\text{Au}_{40}\text{SR}_{24}$ nanoclusters and showed that the two out of the four ($\text{Au}_{24}\text{SR}_{20}$ and $\text{Au}_{40}\text{SR}_{24}$) show significant deviation from the BE/CE parity. We have also added to the manuscript brief discussion on the ability of the model for prediction. Additionally, following the suggestion of the reviewer, we plotted the number of surface core atoms and the shell interactions as a function of $n_c^{-1/3}$ (see Figure below) and we observe that both these factors do not scale well with the $n_c^{-1/3}$. The trend lines display a decent R^2 value, however the lower and higher n_c belie a different functionality.

Action:

- We have added the following discussion to the manuscript following the introduction of the $[\text{Cu}_{25}\text{SR}_{18}]^-$ nanocluster (NC):

“Therefore, we suggest that, at least for this ligand configuration (type and number of ligands), the $[\text{Cu}_{25}\text{SR}_{18}]^-$ cannot be a “magic number” NC. We thus believe that the $[\text{Cu}_{25}\text{SR}_{18}]^-$ serves as a case where the core CE is not balanced with the shell to core BE, ruling out this energetic balance as a simple interfacial effect. Additionally, we have tested our method on four theoretically-predicted Au NCs, the $\text{Au}_{18}\text{SR}_{14}$, $\text{Au}_{20}\text{SR}_{16}$, $\text{Au}_{24}\text{SR}_{20}$, and $\text{Au}_{40}\text{SR}_{24}$, and showed that two ($\text{Au}_{24}\text{SR}_{20}$ and $\text{Au}_{40}\text{SR}_{24}$) out of four exhibit significant deviation from parity, hinting that this energetic balance is sensitive to the actual NC structure and can further screen theoretical NCs predicted with current best practices (see S4).^{11,36–38}”

- In addition, we have added the S4 and following discussion as discussed below (with new core CE definition from response (b) to Reviewer 3):

Figure S4: Parity plot between core CE (kcal/mol) and the shell to core BE as suggested by our developed structure-dependent stability theory. The majority of the values are identical to Figure 1. Additional points include: Ag₂₅SR₁₈⁻ R=CH₃ (a) optimized and (b) experimental structures, respectively, the theoretically predicted (c) Au₂₄SR₂₀ (ref.¹), (d) Au₄₀SR₂₄ (ref.²), (e) Au₁₈SR₁₄ (ref.³), (f) Au₂₀SR₁₆ (ref.⁴), and (g) the Au₁₈SR₁₄ R=C₆H₁₁ optimized structure. The silver arrow indicates the shift of the core CE and shell to core BE during optimization. In the nanocluster (NC) image, the red ball/stick represent the Experimental structure, whereas, the yellow sticks, the Optimized structure.

Comments S4:

For the [Ag₂₅SR₁₈]⁻ NC (S4 (a)-(b)), the geometric reconstruction during optimization is noticeable and it is due to the lack of hydrogen-bonding in R-groups when R=methyl. To additionally verify that the methyl R group substitution does not alter the stability of other Au NCs, we optimized the Au₁₈SR₁₄ with the original R=C₆H₁₁, S4(g), finding a variation of only 0.3 kcal in the core to shell BE while the core CE remained identical to the R=CH₃ structure (S5(g)). For the theoretically predicted Au₁₈SR₁₄, Au₂₀SR₁₆, Au₂₄SR₂₀, and Au₄₀SR₂₄ NC (S5(c)-(f)) several observations can be made. First we note that both the Au₁₈SR₁₄ and Au₂₀SR₁₆ predicted NCs showed energetics matching core to shell BE and core CE, indicating that these structures fit our “structure-dependent stability” model but have not yet been experimentally observed. Next, the Au₂₄SR₂₀ and Au₄₀SR₂₄ predicted NCs are observed to fall well outside the parity line, indicating that although these were previously predicted as stable, they are likely not the actual experimentally synthesized structures. Thus, our structure dependent stability theory can serve as an effective screen for theoretical NC structure predictions.”

- Additionally, the following sentence was added to the discussion section:

“Finally, this novel theory directly relates these thermodynamic stability (energy) contributions to geometrical characteristics of metal cores of the NC, rationalizing for the first time, NC size and shape effects on NC stability and opening new avenues for *in-silico* NC predictions.”

2. The paper advocates a "structural model" with predictive power. In fact a simple $n^{**(-1/3)}$ behavior is found due to the interface effect as discussed above. This model has no predictive power. Why don't the authors use the knowledge and predict here even a few stable compositions? Beyond that comes the structure (is the metal core going to be icosahedral, decahedral, fcc-like, hcp-like, bcc-like... ??) The authors' model says nothing about that, and never can.

Response: The reviewer highlights an important research direction, that of exact structural prediction of ligand-protected nanoclusters. Towards this direction, we have added a substantial discussion and a new figure (Figure 5) as seen in our response to comment 2 of Reviewer 1. With this revision, we demonstrate a general framework for nanocluster structural prediction. Additionally, using the basin-hopping algorithm method previously proposed by Pei. et al. (Pei, Y.; Zeng, X. C. *Nanoscale* **2012**, 4 (14), 4054) and the developed relations highlighted in Figure 3(d) of our manuscript, we have been able to successfully predict the Au₂₅ structure (see below). Along these lines we are actively developing further prediction methodologies but this is beyond the scope of the current work. We believe that our revisions have adequately presented a general framework for nanocluster prediction taking into consideration the developed structure-energy relationships.

Supporting Figure (not for publication): (a) Basin Hopping was applied to the amorphous structure ($n=13$) seen in the upper left of the figure utilizing the Sutton-Chen potential to generate a library of nanocluster cores. After running single point DFT calculations on these generated cores the CE values of each core was plotted on Figure 3(d). At the bottom, different structures of Au_{13} are highlighted that i) fell in the region near the prediction line and ii) formed solid (and not separated) geometric structures. From this analysis, the Au_{13} icosahedral core of the $[Au_{25}SR_{18}]^-$ nanocluster is reproduced, highlighting an example utilizing the relations highlighted in this work for exact structural prediction of the NC core. This metal core prediction can be combined with the stoichiometry relationships (see new Figure 5 from this revision) and our structure-dependent stability relationships (Figure 2 and 3) to predict full thermodynamically stable, ligand-protected nanoclusters.

Action: See actions from comment 2 of Reviewer 1 regarding the nanocluster prediction framework and from comment (b) of Reviewer 3 (including the addition of the updated Figures 2 and 3).

3. Besides just re-discovering a well known interface effects in nanoparticles, the authors make a crucial simplification in the calculations. Omitting the real chemical nature of the ligands and resolving to passivation by SMe only basically neglects all the interactions between the ligands in the ligand layer. The weak organic interactions in the "R-part" of the ligand layer count to several electron volts for tens of ligand and hundreds of carbon atoms and thus provide a significantly increased stabilization effect. The effect is so important that actually there are examples in the literature where the same cluster (same composition) can have different core atomic structure depending on the bulkiness of the ligand. One of the current trends is actually to try to understand the ligand layer structure and the weak interactions in the ligand layer by spectroscopic and computational methods (see, e.g., Nature Comm 7, 10401, 2016).

Response: We agree with the reviewer that the chemical environment imposed by the ligands can have a significant effect on the final nanoparticle (cluster) morphology and that a thorough study of the ligand effects is needed. We have highlighted this fact when examining the $[Ag_{25}SR_{18}]^-$ cluster in the revised SI 4 and Figure 2, where the full, bulky ligands were needed to accurately capture the shell to core interactions. For all other clusters in this study, however, we did not observe any significant geometric

restructuring at the interface between the cores and shells during optimization. As an example of this fact we have highlighted how the Au₁₈SR₁₄ cluster shows little to no energetic difference (BE and CE) for the case when the full ligand is used or not. In fact, although we agree with the reviewer that the ligand-ligand (van-der-Waals) interactions can become important holistically, the energetic analysis in our work (CE or BE) has been made per Au atom. As a result, the metal-metal interactions in the cores and shells of the nanoclusters are significantly stronger than the ligand-ligand dispersion interactions (see: Reimers, J. R.; Ford, M. J.; Halder, A.; Ulstrup, J.; Hush, N. S. *Proc. Natl. Acad. Sci. U. S. A.* 2016, 113 (11), E1424-33.). For this reason, our model successfully captures to a large degree the energetics of the nanocluster. Beyond these facts, the core and shell structures of these precise nanoclusters have been shown to be virtually identical experimentally for a wide variety of different ligands in other studies, highlighting the importance of core and shell structure in the cluster stability (see: Yuan, X.; Goswami, N.; Mathews, I.; Yu, Y.; Xie, J. *Nano Res.* 2015, 8 (11), 3488.)

Action: We added the following discussion to the section where the ligand substitution with methyl-groups is outlined:

“Toward understanding ligand impact we highlight that experimentally, the [Au₂₅SR₁₈]⁻ NC has been shown to be stable for a wide variety of ligands,³⁹ and was successfully synthesized even with small, ethyl R groups.⁴⁰ Therefore, the exceptional structural stability of [Au₂₅SR₁₈]⁻ NC seems to be experimentally independent of the ligand type, highlighting the importance of metal structure and Au/S stoichiometry in determining stable NCs. For NC structures investigated in this work interactions at the interface between their core and shell regions should be to a large degree unaffected by the ligand selection⁴³ (see S4 where Au₁₈SR₁₄ optimization with full ligands resulted to minor shift and SI file for detailed analysis on exceptions). In addition, metal-metal interactions at the interface are energetically far stronger than the ligand-ligand interactions and capture the core-shell and the relative NC stability. However, enhanced ligand-ligand (R-group) interactions can impact the overall NC stability and associated physicochemical properties as seen in several other recent works.^{41,42”}

See actions in response to Reviewer 2 comment 1 for updated S4.

4. The use of the dielectric model for WATER as a solvent is meaningless. The clusters are computed with the SMe ligand layer, which (if this ligand really could be used in the actual synthesis), would make the cluster material NON-water-soluble.

Response: We thank the reviewer for this comment. The purpose of using water (most commonly used solvent) was to demonstrate that the relationships work also when accounting for polar media (in addition to the gas phase). In order to address the reviewer’s concern we have repeated the same analysis simulating a commonly-used nonpolar solvent (dichloromethane (DCM), epsilon=8.93). We again found no significant deviation from the parity.

Action: We have added S5(a) with DCM figure (Below) and added the following discussion to the paper (BE and CE values reported follow updated definitions from response (b) to Reviewer 3):

“To further prove that this structural thermodynamic stabilization is a general behavior and originates solely from the energy balance between the core and the shell of the NCs we analyzed CE and BE in the presence of the common⁸ dichloromethane (DCM) (see S5(a)) and water (S5(b)) solvents.”

Figure S5: Parity plot between core CE (kcal/mol) and the shell to core BE including (a) dichloromethane ($\epsilon=8.93$) and (b) water ($\epsilon=78.46$) solvent effects, using the COSMO implicit solvation model.

Comment S5: Similar to Figure 2, the parity between shell to core BE and core CE holds. Very slight shifts of the shell to core BE are introduced by the presence of the solvent without affecting the overall trends.

5. The authors ignore very basic chemical facts when discussing other metals than gold. Some years ago, most of the community thought that only gold can be made stable in nanoparticle form with thiolate or phosphine ligands. Since then several groups have discovered the right chemistry to stabilize a number of silver nanoclusters that have been crystallographically solved to atomic precision (and this referee happens to know that new breakthroughs in that front are going to be published soon). The authors throw a case of the copper cluster $Cu_{25}(SMe)_{18}(-)$, computed in the hypothetical structure of Au_{25} , as a "negative prediction". Of course, most likely the cluster will not be stable in this structure. It simply might find a better, global minimum, with a suitable (bulky) ligand layer that protects the core from oxidation. Oxidation of copper and the chemistry to protect from it are the challenges (back to point 3).

Response: We agree with the reviewer that oxidation of copper is likely responsible for the relative lack of reported copper nanoclusters. We believe, however, that such effects are beyond the scope of our work which focuses entirely on the stability of nanoclusters. The copper particle provides an interesting counter example to the comment 1 of this reviewer, where we show that this methodology does not simply reveal an interface effect. Additionally, this Cu nanocluster would be predicted "stable" by the superatom theory and therefore highlights another advantage of this new model over previous theories.

Action: We have added the following discussion to the manuscript: "While the challenge with synthesizing Cu NCs is largely tied to the persistence of the Cu(I) state³⁵, our calculation imposes the

ideal experimental case where the Cu in $[\text{Cu}_{25}\text{SR}_{18}]^-$ remains Cu(0). Therefore, we suggest that, at least for this ligand configuration (type and number of ligands), the $[\text{Cu}_{25}\text{SR}_{18}]^-$ cannot be a “magic number” NC. We thus believe the $[\text{Cu}_{25}\text{SR}_{18}]^-$ serves a case where the core CE is not balanced with the shell to core BE, ruling out this energetic balance as a simple interfacial effect.”

6. From the title one gets impression that an interface energy analysis would have been made also for silver nanoclusters protected by thiolates. Several crystallographically solved structures are available in the literature! However, this is not done.

Response: This analysis was already performed for the $[\text{Ag}_{25}\text{SR}_{18}]^-$ cluster and appeared in Figure 2 and the SI of the originally submitted manuscript.

7. Comparison of metal core energies to energies of free gas phase metal clusters with the same nuclearity is without a merit. Structures of free gold clusters in gas phase was a hot topic 10-15 years ago. The authors are welcome to review the extensive literature from that time. Structures seen in the metal cores of thiolated gold nanoclusters have little or nothing to do with structures of free gas phase gold clusters of the same nuclearity. This is a fact.

Response: We have highlighted the cohesive energy difference between the gas phase and NC core structures to rationalize the impact on the core cluster CE from the presence of the shell. We have demonstrated how the CE comparison at the same nuclearity highlights the predictive power of the model as shown in our response to the second comment of reviewer 2.

8. A minor point: when talking about the "divide and protect" idea, the authors cite ref. 8. The correct reference is #15 (just look at the title of ref. 15 !).

Response: We thank the reviewer for this minor correction and we have addressed this in our revised manuscript.

Reviewer #3 (Remarks to the Author):

This paper reports on computational work used for deriving stability descriptors for ligand-protected gold clusters. The subject is interesting, the approach and the results are original, but the paper itself is not convincing. I have some major concerns at this stage so that I cannot recommend publication.

Response: We thank reviewer 3 for their valuable insights and suggestions for improving our work and for their evaluation of the work as original and interesting.

Major points

a) The key quantities BE and CE are not clearly defined. For what concerns BE, in Equation (1) it is written that n_{shell} is the number of metal atoms of the shell interacting with the core. This looks quite arbitrary, and in the text it is not explained how these interacting atoms are chosen. Some kind of explanation is given in the comments of the supplementary material, which however do not completely convince about the arbitrariness of the definition. For what concerns CE, there are several unclear points. First of all, is $E_{metalatoms}$ the total energy of an isolated Au atom? If yes, why multiply it by $n_{metalatoms}$? Since core energy is concerned, this should be multiplied by the number of core atoms n_{core} , which should also appear in the denominator of Equation (2).

Response: To address this arbitrariness of the definition of the number of shell metal atoms interacting with the cores we have created a more systematic way of identifying the number of shell contacts as outlined below (see action), which is consistent with results shown previously. The $E_{metalatoms}$ as the reviewer has identified actually represents the total electronic energy of an isolated Au atom and should therefore be written $E_{metalatom}$. As a result, the cohesive energy of the core is: $CE = \frac{E_{Cluster} - n_c E_{Metal Atom}}{n_c}$.

We thank the reviewer for pointing this unclear detail.

Actions:

- We have added the following discussion to the methods section:

“Two methods were used to identify if Au atoms were “core” or “shell”, that of natural bond orbital (NBO) charge analysis and that of measuring S-contacts of Au atoms in the structure, where the shell Au atoms have exactly 2 bonded sulfur groups (see S1 for example of Au₂₀ determination).⁶⁶ These two methods were in perfect agreement over all NCs.”

“ $n_{shellInt}$ is largely dictated by the number of shell Au atoms in contact with the surface of the cores (less than 4 Å from the nearest core Au atom) because metal-metal bonds dominate the shell to core BE (see S3). Beyond shell Au contacts to the cores, SR groups that are not bound to any shell Au but are bound directly to core Au represent a direct shell-to-core contact and thus are also included in $n_{shellInt}$ (see S1 for details surrounding the calculation of $n_{shellInt}$).”

- We also replaced S1 with the following:

Figure S1: Optimized structure of $\text{Au}_{20}\text{SR}_{16}$, $\text{R}=\text{CH}_3$. Top: charge analysis, where red tints indicate negative and blue positive charges, respectively (darkest red=-0.76, darkest blue=+0.23). Dark blue metal atoms are counted as shell. Similarly, Au atom (a) shows bonds (highlighted yellow) to two sulfurs (making it a shell Au), while Au atom (b) only shows one bond to a sulfur (making it a core Au). Bottom: HOMO orbital structure. Highlighted (green) Au atoms show more bonding character with bridging shell metal atom (indicated with black arrow) and therefore are counted as non-interacting shell metal atoms.

“

- and added Comment S1 in the supporting information file:

“The metal atoms were first determined as core vs. shell by examining their NBO charge state. Metal atoms with more than 0.2 charge were identified as shell (indicating partially cationic Au atoms). This 0.2 charge threshold was established based on the charge of the shell Au atoms in all of the structures (via same analysis as in S1). For this same core vs. shell determination, alternatively, metal atoms that were coordinated to 2 sulfurs were assigned as shell metal atoms and all other metal atoms were assigned as core. These two methods produced identical results. To determine the number of interacting metal atoms we identified the distance between the shell metal atoms and their nearest core atoms. Assuming an interaction distance cutoff for bonding at approximately 2.5 times the van der Waals radii for the Au metals (4 Å), the interacting (non-interacting) metals can simply be counted by the number of shell metal atoms with a minimum shell-core distance less (larger) than this cutoff. This automated process exactly results in the energy balance shown in Figure 2 for every NC (CE=shell to core BE) with the exception of Au₂₀(SR)₁₆ (and the negative test [Cu₂₅(SR)₁₈]⁻ NC). For the Au₂₀(SR)₁₆ NC, further examination of the HOMO (electronic) orbital structure indicated primary bonding for 2 of the atoms identified as interacting shell with another shell Au atom as shown in S1. This indicated that these atoms could more accurately be represented as non-interacting shell atoms despite their close proximity to one of the core metal atoms. For the charged systems ([Au₂₅SR₁₈]⁻, [Cu₂₅SR₁₈]⁻, and [Ag₂₅SR₁₈]⁻) we performed vertical electron affinity calculations between the separated core and shell regions to identify where the negative charge will be located. In all cases, the shell region showed a higher electron affinity and the electron was attributed to the shell in the charged systems.”

b) The derivation of these stability descriptors is not convincingly supported by some line of reasoning. Some kind of thermodynamic reasoning is given in the supplementary material. In my opinion, the derivation of the descriptors should be supported by some physical line of reasoning besides (possible) numerical evidence. The arguments in the supplementary material are only sketched and not convincing. The authors should clearly indicate what are the initial and final states of their thermodynamic reasoning, taking also into account that the core covered by shell atoms in the process, and whether this affects the balance of eq. (5).

Response: We thank the reviewer for pointing us to look at cohesion in the presence of the shell. Along this line we derived a new thermodynamic model that proves why the CE/BE energy balance should hold for thermodynamically stable nanoclusters. We calculated the CE in the presence of the shell for all the nanoclusters and we showed a perfect correlation with the CE of just the gas phase cores (without the shell). As a result, we have updated (and added the Au₃₀S(SR)₁₈ following reviewer 1 comment 1) Figures 1, 2, and 3 and Table S5 of the manuscript and added a new Figure, S6 (showing the parity between the CE in the presence of shell and in its absence). Additionally, the results presented in response to reviewer 2 comments 1 and 4 were based on this new analysis.

Actions:

- We have edited the red sections of text below in the thermodynamic analysis part of the SI file:

“Thermodynamic analysis rationalizing the NC stability in the “*structure-dependent stability*” model.

The stability descriptors of core CE and shell to core BE can also be linked to thermodynamic parameters. Since we are separating the NC in two distinct phases (core and shell, according to *divide and protect*) which are in direct contact, and in order to achieve chemical equilibrium, the partial molar Gibbs free energy (chemical potential (μ)) of the two phases should be equal (so $\Delta\mu(\text{NC})=0$). Towards this end we

select a thermodynamic reference state that corresponds to the solution in the Brust-Schiffrin synthesis immediately following the addition of the reducing agent, consisting of solvated M^0 and “staple groups”, $SR-(M-SR)_n$, that then, self-assemble to form the NC core and shell regions, respectively. For example, assume a solution where immediately following the addition of the reductant 23 Au^0 atoms exist along with 6 $SR-Au-SR-Au-SR$ and 3 $SR-Au-SR$ groups, in addition to excess thiol and solvent. This would correspond to the initial state of the thermodynamic argument while the final state would be the assembled $Au_{38}SR_{24}$ cluster in the same solution. We can assume that the difference of the partial molar entropy (s) of the M^0 and “staple group” between the reference solution and the NC phases are equivalent.⁶ In addition, the core CE in the presence of the shell can largely represent the partial molar enthalpy (h) of the Au atoms in the core relative to the reference solution phase, as the electronic energy will dominate the h values in a constant volume, liquid phase reaction. To make the analysis of the core CE in the presence of the shell we can rely on the core CE and shell to core BE already observed.

$$Core\ CE\ (with\ shell) = \frac{(E_{Full\ Cluster} - n_c * E_{Metal\ Atom} - E_{ShellInt})}{(n_c + n_{ShellInt})} \quad (1)$$

Where E_x is the electronic energy of species X, n_c is the number of metal atoms in the core, and $n_{ShellInt}$ is the number of interactions between the shell and core. This equation can then be rewritten as:

$$Core\ CE\ (with\ shell) = \frac{(E_{Full\ cluster} + n_c * CE_{Core} - E_{Core} - E_{Shell})}{(n_c + n_{ShellInt})} \quad (2)$$

Where $CE_{core}=(E_{core}-n_c * E_{Metal\ Atom})/n_c$, representing the atomization energy for the isolated gas-phase core as defined in the methods section of the manuscript. This equation can then be rearranged as:

$$Core\ CE\ (with\ shell) = \frac{(n_c * CE_{Core} + n_{ShellInt} * BE_{shell-to-core})}{(n_c + n_{ShellInt})} \quad (3)$$

Where $BE_{shell-to-core}=(E_{NC}-E_{core}-E_{shell})/n_{ShellInt}$, as defined in the methods section. Thus, the cohesive energy of the metal atoms in the core in the presence of the shell can be viewed as a weighted average of the isolated core CE and shell to core BEs. Finally, the shell to core BE likewise is treated as the h of the core-binding shell M atoms relative to the reference solution phase, considering interactions between staple groups are known to be very weak relative to their interactions with the core M^0 atoms.⁷ These assumptions are summarized as:

$$\Delta\mu_{M\ Core}^{Solution-NP} = \Delta h_{M\ Core} - T\Delta s_{M\ Core} \approx Core\ CE(with\ shell) - T\Delta s_{M\ Core} \quad (4)$$

$$\Delta\mu_{M\ Shell}^{Solution-NP} = \Delta h_{M\ Shell} - T\Delta s_{M\ Shell} \approx BE(shell - to - core) - T\Delta s_{M\ Core} \quad (5)$$

Where μ is chemical potential, h is partial molar enthalpy, s is partial molar entropy, T is temperature, and Solution-NC indicates the difference between the reference solution and NC atoms. Thus, the chemical potential difference between the surface and core metal atoms is given as:

$$\Delta\mu(NP) = 0 = \Delta\mu_{M\ Shell}^{Solution-NP} - \Delta\mu_{M\ Core}^{Solution-NP} \approx BE(shell - to - core) - Core\ CE\ (with\ shell) \quad (6)$$

Which indicates that for the stable NCs we have identified, this difference in chemical potential will be equal to zero, highlighting a balance of chemical potentials at this core-shell interface. This thermodynamic analysis helps rationalize the lack of temperature-dependence in the stability of NCs in temperature regimes where enthalpic dominate entropic contributions. In addition, it demonstrates why our developed “structure-dependent stability” model is a valid thermodynamic model for NC stability.”

- Additionally, we have updated the methods sections with the following equations:

“The shell to core BE is defined as:

$$\text{Shell to Core BE} = \frac{E_{Full\ Cluster} - E_{Shell} - E_{Core}}{n_{ShellInt}} \quad (1)$$

where E_x = electronic energy of group X and $n_{ShellInt}$ = number of shell contacts interacting with the core (See Figures 1&2).”

“The core CE is defined as:

$$\text{Core CE} = \frac{E_{Full\ Cluster} - n_c E_{Metal\ Atom} - E_{Shell}}{n_c + n_{ShellInt}} \quad (2)$$

where n_c = number of metal atoms contained in the core structures (and E is the total electronic energy).”

“For the gas phase minimum energy clusters and NC core structures the CE is defined:

$$CE = \frac{E_{Cluster} - n_c E_{Metal\ Atom}}{n_c} \quad (3)”$$

- We added discussion in the text surrounding the parity between Gas Phase CE and Core CE values and S6 as evidence:

“Interestingly, the gas phase CE (equation 3) is roughly equivalent to the core CE calculated with the presence of the shells in the NCs (see equation 2 and S6). Therefore, the gas phase CEs of the NC cores, accurately represent the stability of the cores in the NC (presence of shell), and can be directly compared with the gas phase global minimum energy structures, in Figure 3(d).”

Figure S6: Parity plot between core CE (kcal/mol) and the gas phase core CE, indicating remarkable parity between the gas phase and shell-influenced CE values.

- We have replaced Figures 1,2, 3(a-d) with the new Core CE and Shell to Core BE values to illustrate the parity between the Core CE and Gas Phase Core CE as shown below:

Figure 1: Optimized geometries of the experimentally-synthesized (a) Au₁₈SR₁₄²⁷ (b) Au₂₀SR₁₆²³ (c) Au₂₄SR₂₀²⁸ (d) Au₂₈SR₂₀³⁰ (e) Au₃₀S(SR)₁₈¹⁹ (f) Au₃₆SR₂₄³¹ (g) Au₃₈SR₂₄q³² (h) Au₃₈SR₂₄t⁶¹ (i) Au₁₀₂SR₄₄³³. n_c represents the number of core metal atoms while n_{ShellInt} represents the number of shell to core interactions. Ligands (S-CH₃) are shown in stick representation while core and shell atoms, in ball and stick, and have been colored yellow and blue, respectively. In (b) and (c), shell Au atoms which do not interact with the core have been colored red and are shown in stick representation while in (a) and (i) shell Au atoms which were previously identified as core are colored darker. In (e) and (h) shell sulfurs which are not directly bound to a shell Au atom are shown as brown balls.

Figure 2: Parity plot between core CE (kcal/mol) and the shell to core BE. The corresponding structures of the $Au_n(SR)_m$ NCs are presented in Figure 1 except from the optimized structures of (a) $[Au_{25}SR_{18}]^-$, (b) $[Cu_{25}SR_{18}]^-$, and (c) $[Ag_{25}(SPhMe_2)_{18}]^-$ NCs, which are shown as insets in the graph. For (a)-(c) $n_c=13$ metal atoms (Au/Cu/Ag) and $n_{shellm}=12$ as in Figure 1. The shell metal atoms are shown in blue, whereas, the Cu and Ag core metal atoms are shown in red and green, respectively. Here, all the Au and Ag NCs reported have been experimentally determined. The Cu NC structure is hypothetical, optimized from the Au NC analogous structure (a).

Figure 3: (a) Core CE and shell to core BE vs. $n_c^{-1/3}$ (number of core metal atoms) for cores of thermodynamically stable “magic number” Au NCs (b) Core CE and shell to core BE vs. Average Coordination numbers (CNs) for cores of Au NCs (c) Shell to core BE and Average Core CN vs. the ratio of total Au atoms and S atoms in the shells and (d) Global minima gas phase Au clusters and cores of Au NCs. From Figures 1 and 2, the Au NC cores contain: $Au_{18}SR_{14}=8$, $Au_{20}SR_{16}=7$, $Au_{24}SR_{20}=8$, $[Au_{25}SR_{18}]^-=13$, $Au_{28}SR_{14}=14$, $Au_{30}S(SR)_{14}=17$, $Au_{36}SR_{24}=20$, $Au_{38}SR_{24}q, t=23$, and $Au_{102}SR_{44}=77$ Au atoms, respectively.

Table S1: Core CE (kcal/mol) and shell to core BE (kcal/mol) from BP-86^{6,7}, PBE⁸ and BLYP^{7,9} from single point energy calculations on the BP-86 optimized [Au₂₅SR₁₈]⁻ structure. There is a tight match between the core CE and shell to core BE for the BLYP, BP-86, and PBE methods, which are all GGA methods.

Method	Core CE (kcal/mol)	Shell to Core BE (kcal/mol)
ri-BP-86	-37.82	-37.40
ri-PBE	-38.95	-39.29
ri-BLYP	-31.0	-28.57

Minor points- The authors do not consider van der Waals corrections. Some checks including them should be made.

Reply: We thank the reviewer for bringing up this choice of methods, however, we have purposely chosen to not include in our calculations Van Der Waals corrections, due to their overestimation of the bond strength at the interface of interest.

(1) Andersson, M. P. J. *Theor. Chem.* 2013, 2013, 1–9.

(2) Reimers, J. R.; Ford, M. J.; Halder, A.; Ulstrup, J.; Hush, N. S. *Proc. Natl. Acad. Sci. U. S. A.* 2016, 113 (11), E1424-33.

Action: We added a sentence to methods section detailing lack of van der waals corrections.

“We did not include van der waals corrections in our calculations as they tend to overestimate Au-Au bonding at the interface of Au-thiolate layers.^{2,3}”

- Figure 2c looks rather confusing. Why the values of CN are in decreasing order? If these values are put in increasing order, it is immediately clear that high CN correspond to strong BE.

Reply: We have chosen this representation to highlight, how the two lines trend opposite of intuition with high CN showing strong BE.

Reviewers' comments:

Reviewer #1 (Remarks to the Author):

The authors have revised the manuscript to address the majority of my comments.

One major issue remains: "Structure dependent stability".

The authors admit that they do not mean structure, by adding, "We have added the following clarification sentence to the end of the introduction: "In this context we define structure as composition (Au vs. S content) in addition to NC size and shape (morphology)."

With such poor justification to use the word, "Structure-dependent stability", I would not use the word "Structure" in the title.

I would suggest using either "Thermodynamic stability" or "Composition-dependent stability" taken from the abstract and the authors response, respectively.

Minor point: It would be interesting to check the dependence on the larger clusters, such as DOI: 10.1021/nn501970v, and Au₂₄₆(SC₆H₄CH₃)₈₀, perhaps in a future study.

Reviewer #3 (Remarks to the Author):

Reply is convincing.

I recommend publication

Reviewer #4 (Remarks to the Author):

The present manuscript present a theoretical analysis of the stability of ligand-protected metal nanoclusters in the size range M18-M102. These species have a metal core protected by a ligand shell, often metal/thiolate shell, and are a popular field of research. The main results of the present work is that the plot of shell-to-core vs core cohesion energies is linear in this size range. This result is interesting. However (being an adjudicative assessment) I agree with previous reviewers that this linear trend is basically a consequence of scaling size relationships which are well known in cluster science. The most noteworthy feature is that the intercept is zero and the slope unity, which suggests a good balance of approximately equal strengths of Au-Au and Au-S bonding in this size range, something which is possibly not true for Cu clusters. Apart from this, however, this result does not clarify whether there is any difference in the stability of the investigated species or better which species are the most stable. Moreover, it can be noted that a significant deviation occurs for Au₁₀₂, which is the largest cluster, thus either Au₁₀₂ is not advocated as particularly stable or the linear trend is only valid in a limited size range due to coincidental size scaling balance. Finally, as underlined by previous reviewers but not convincingly responded, ligand-ligand interactions have been shown to be crucial for the stability of these species and neglecting them does not explain experimental findings.

For these reasons I don't recommend publication of the present manuscript in Nature Communications.

PS Note that the literature citation is also incomplete. For example, electronic and charge analysis has

been first performed in Ref. DOI: 10.1021/nn800268w (later work by Zeng's group are quoted). Ref. DOI: 10.1021/ja101083v is quoted to draw that the metal-metal interactions in the cores and shells of the nanoclusters are significantly stronger than the ligand-ligand dispersion interactions, which is true but hides the fact that residual effects are crucial to determine the differential stability of these species, outside trends with scaling size. Other stability analysis such as DOI: 10.1039/C2NR30501A (with the correlation between Au-S and shell to core binding strength) and DOI: 10.1021/ja507738e (with energy decomposition analysis) are not mentioned whereas this work uses techniques there previously proposed.

We would like to thank all three reviewers for their time and effort to evaluate our revised manuscript. Two out of three reviewers (specifically Reviewers 1 and 3) were satisfied by our significant revision and recommended publication in Nature Communications. Although the third reviewer (Reviewer 4) finds the main results of our study to be interesting (“The main results of the present work is that the plot of shell-to-core vs core cohesion energies is linear in this size range. This result is interesting.”), he/she feels that our results are a consequence of scaling size relationships on clusters. We have revised our manuscript by addressing all the comments of Reviewer 4. Specific responses and actions are listed in this “point-by-point response” file. All changes are highlighted in red in the revised manuscript and supplementary information file.

Reviewers' comments:

Reviewer #1 (Remarks to the Author):

The authors have revised the manuscript to address the majority of my comments.

One major issue remains: "Structure dependent stability".

The authors admit that they do not mean structure, by adding, "We have added the following clarification sentence to the end of the introduction: “In this context we define structure as composition (Au vs. S content) in addition to NC size and shape (morphology).”

With such poor justification to use the word, "Structure-dependent stability", I would not use the word "Structure" in the title.

I would suggest using either "Thermodynamic stability" or "Composition-dependent stability" taken from the abstract and the authors response, respectively.

Response: We appreciate the reviewer’s perspective on the title for this work.

Action: We have updated the title of the manuscript to “Thermodynamic Stability of Ligand-Protected Metal Nanoclusters.”

Minor point: It would be interesting to check the dependence on the larger clusters, such as DOI: 10.1021/nn501970v, and Au₂₄₆(SC₆H₄CH₃)₈₀, perhaps in a future study.

Response: We thank the reviewer for bringing into our attention this work. It can definitely be a system for future study utilizing more efficient computational packages and DFT algorithms, as well as, large supercomputing facilities.

Reviewer #3 (Remarks to the Author):

Reply is convincing.

I recommend publication

Response: We thank the reviewer for the positive response.

Reviewer #4 (Remarks to the Author):

The present manuscript present a theoretical analysis of the stability of ligand-protected metal nanoclusters in the size range M18-M102. These species have a metal core protected by a ligand shell, often metal/thiolate shell, and are a popular field of research. The main results of the present work is that the plot of shell-to-core vs core cohesion energies is linear in this size range. This result is interesting.

Response: We thank the reviewer for recognizing that the main result of our study is interesting in a popular field of research.

However, (being an adjudicative assessment) I agree with previous reviewers that this linear trend is basically a consequence of scaling size relationships which are well known in cluster science.

Response: To further address the concern of simple size/scaling relations (beyond our detailed responses to the comments 1 of reviewer 2 and comment 2 of reviewer 1 in our previous revision) we have performed additional analysis calculating the surface area of the cores of the nanoclusters (NCs). The surface area of the cores was calculated using the Shrake-Rupley algorithm.¹ It is known that the surface area is inherent in NC scaling relations as they are largely based on trends of surface area to volume ratio.² Based on these relationships one can predict the number of thiolate groups needed to protect a NC of specific size. We have actually used these relationships in this work to predict the stoichiometry of thiolate protected Au NCs (see Figure 5). Our introduced model however, shows a previously undiscovered perfect energy balance between the cohesive energy (CE) of the metal core and the shell to core binding energy (BE). This is not a result of scaling relationships. Assuming that linear scaling size relations dominated in our model, then we would expect that the ratio of the surface area of the NC cores to the number of shell to core contacts (n_{ShellInt} , which determine the BE) to be constant, as these contacts would be expected to interact uniformly over this area at the surface cores. Upon modeling this ratio as a constant and starting from the smallest NC ($\text{Au}_{18}\text{SR}_{14}$), we see below (Supporting Figure) that this linear scaling assumption (red line) fails to capture several NCs (large deviations from the red line), which becomes completely unrealistic (no trend) for the larger $\text{Au}_{102}\text{SR}_{44}$ NC. Additionally, there are deviations from the number of shell to core interactions for cluster cores exhibiting almost identical surface areas, such as the $\text{Au}_{20}\text{SR}_{16}$ and $\text{Au}_{24}\text{SR}_{20}$ NCs, as well as, deviations in the surface area of cores on NCs that exhibit the exact same number of shell interactions, such as the $\text{Au}_{24}\text{SR}_{20}$ and $[\text{Au}_{25}\text{SR}_{18}]^-$ NC cores. Therefore, the perfect energy balance between the core CE and the shell BE is not due to linear trends and this is the reason why the Au/S ratio on the shell of the NCs changes with size (see Figure 3c of our manuscript) and does not remain the same at every NC size. The Au/S ratio on the NC shell modulates the CE and BE energy balance.

Furthermore, if this energetic balance was simply a result of a scaling relation our model would not be able to differentiate between molecular isomers. In our response to comment1 of reviewer 2 (previous revision) we had highlighted the theoretical molecular isomer of $\text{Au}_{24}\text{SR}_{20}$ for which

the CE-BE balance was not maintained. The model proposed in this work therefore has been shown to move beyond simple scaling relations, and has been shown to tie the thermodynamic stability of the experimental NCs to their structure.

1. Shrake, A. & Rupley, J. A. Environment and exposure to solvent of protein atoms. Lysozyme and insulin. *J. Mol. Biol.* 79, 361–371 (1973).
2. Dass, A. Nano-scaling law: geometric foundation of thiolated gold nanomolecules. *Nanoscale* 4, 2260 (2012).

Supporting Figure (not for publication): Surface area of Cores (Å²) vs. n_{ShellInt} for all experimental Au NC structures reported in the manuscript. The red line shows a simple linear scaling of Surface Area/ n_{ShellInt} relationship based on the data of Au₁₈SR₁₄ NC. There are significant deviations from the red line (see pronounced case of Au₁₀₂SR₄₄).

The most noteworthy feature is that the intercept is zero and the slope unity, which suggests a good balance of approximately equal strengths of Au-Au and Au-S bonding in this size range, something which is possibly not true for Cu clusters.

Response: Figure 2, which is the central figure of the manuscript and that the reviewer is referred to, is a parity plot between the core CE and the shell BE and there is not any equation fitted to the data. This highlights the importance of the CE-BE perfect energy balance that our model introduces. It appears that the reviewer may have misunderstood this important aspect of our work, which may have resulted in underestimating the significance of our results. We hope that our responses and revisions clarify all these aspects.

Apart from this, however, this result does not clarify whether there is any difference in the stability of the investigated species or better which species are the most stable. Moreover, it can

be noted that a significant deviation occurs for Au102, which is the largest cluster, thus either Au102 is not advocated as particularly stable or the linear trend is only valid in a limited size range due to coincidental size scaling balance.

Response: We thank the reviewer for this comment. It should be noted again that Figure 2 is not a linear trend but a parity plot. However, to quantify the distance from the parity line as a descriptor for the NC stability, we have performed a linear regression with superimposed 95% confidence and prediction statistical bands on only the experimental NC structures. While the Au₁₀₂SR₄₄ falls well within the 95% prediction band (not surprising as it is included in the regression), the reported theoretically predicted structures (and not experimentally synthesized), however, including the [Cu₂₅SR₁₈]⁻, Au₂₄SR₂₀, and Au₄₀SR₂₄ NCs fall well outside the 95% prediction band. Thus, the deviation of the Au₁₀₂SR₄₄ is demonstrably less with respect to the deviation of other theoretically predicted NC structures and the 95% prediction band may serve as a cutoff for “stable vs. not” NCs. To further illustrate this 95% prediction band as a functional cutoff, we have fully optimized the hypothetical Ag₁₈SR₁₄, Cu₁₈SR₁₄, Ag₃₈SR₂₄, and Cu₃₈SR₂₄ NCs by substituting the corresponding Au NCs with Cu and Ag. These new theoretical structures are all shown to fall outside the 95% prediction band, further validating that the 95% prediction band serves as an effective cutoff for stability, which the Au₁₀₂SR₄₄ falls well within. In addition, it should be noticed that these new data show significant CE-BE imbalances (e.g. Cu₃₈SR₂₄), further demonstrating that our proposed model is not simply a scaling relationship. Finally, we note that we have not made any claim that our model can discern the relative stability between the experimentally-synthesizable NCs, but rather we introduce a model that rationalizes why specific NC structures can be synthesized or not.

Actions:

We have added the new Figure S4 below to the SI file:

Figure S4: Parity plot between core CE (kcal/mol) and the shell to core BE with only the experimentally-isolated structures. A linear regression along with 95% Confidence and Prediction statistical bands have been superimposed in blue.

We have added a new Figure S5 (data from the previous S4 with a few additional points) and comment S5 in the SI file:

Figure S5: Parity plot between core CE (kcal/mol) and the shell to core BE with identical 95% Confidence and Prediction bands as in Figure S4. Additional points to Figure 2 of the manuscript include: i) theoretically predicted Au nanoclusters (NCs) (a) $Au_{24}SR_{20}$ (ref.¹), (b) $Au_{18}SR_{14}$ (ref.²), (c) $Au_{40}SR_{24}$ (ref.³), and (d) $Au_{20}SR_{16}$ (ref.⁴) and ii) NCs of different metals (*) generated and optimized from their analogous experimental Au NC structures.

Comments S5: All the experimental NCs fall within the 95% Prediction Bands from the linear regression while all the predicted NCs structures reported, except the $Au_{18}SR_{14}$ and $Au_{20}SR_{16}$ NCs, fall outside the 95% Prediction Bands. We note that both the $Au_{18}SR_{14}$ and $Au_{20}SR_{16}$ predicted NCs showed energetics matching core to shell BE and core CE, indicating that these structures fit our “structure-dependent stability” model but have not yet been experimentally synthesized. Thus, the 95% Prediction bands can be used to distinguish between non-stable and potentially stable NCs.

We have also revised our discussion in the manuscript focused on cluster stability and theoretical predictions:

“To develop a quantitative boundary between synthesizable and non-synthesizable NCs we performed a linear regression on all the experimentally-synthesized NCs with 95% confidence and superimposed the prediction bands (Figure S4). To explore the effectiveness of the 95% confidence and prediction bands in distinguishing between non-stable and stable NCs we optimized additional hypothetical NCs. Beyond the hypothetical $[\text{Cu}_{25}\text{SR}_{18}]^-$ NC, we investigated the $\text{Ag}_{18}\text{SR}_{14}$, $\text{Cu}_{18}\text{SR}_{14}$, $\text{Ag}_{38}\text{SR}_{24\text{q}}$, and $\text{Cu}_{38}\text{SR}_{24\text{q}}$ theoretical NCs generated directly from their corresponding Au NC analogs. We found that they exhibit CE and BE values that deviate beyond the 95% prediction band (Figure S5). Additionally, we have tested our method on four theoretically-predicted Au NCs, the $\text{Au}_{18}\text{SR}_{14}$, $\text{Au}_{20}\text{SR}_{16}$, $\text{Au}_{24}\text{SR}_{20}$, and $\text{Au}_{40}\text{SR}_{24}$, and showed that two ($\text{Au}_{24}\text{SR}_{20}$ and $\text{Au}_{40}\text{SR}_{24}$) out of the four exhibit similar deviation from parity as the theoretical Cu NCs, whereas, the $\text{Au}_{18}\text{SR}_{14}$ and $\text{Au}_{20}\text{SR}_{16}$ NCs exhibit the CE and BE energy balance. Therefore, this energetic balance is sensitive to the actual NC structure and the 95% prediction bands can further be used as cutoffs to screen theoretical NCs predicted with current best practices (Figure S5).”

Finally, as underlined by previous reviewers but not convincingly responded, ligand-ligand interactions have been shown to be crucial for the stability of these species and neglecting them does not explain experimental findings.

Response: We acknowledge that in general ligand effects are important in metal nanoparticle growth. However, we believe that our model captures to a large extent the experimental stability (synthesizability) regardless of the treatment with full ligands. As we highlighted in our previous response, our model focuses on strong metal-metal bond interactions at the core and shell regions of the NCs. To further demonstrate that our model on Au NCs is robust against ligand treatments, we have performed an additional calculation with the $[\text{Au}_{25}\text{SR}_{18}]^-$ NC accounting for the full PhenylEthaneethiol (see ref. 29) ligands. Like the previously-reported $\text{Au}_{18}\text{SR}_{14}$ ($\text{R}=\text{C}_6\text{H}_{11}$), full-ligand NC, we see that the full ligands impart little to no difference compared to using methyl-thiolates on the values of the BE and CE of these NCs. This helps reinforce the concept that ligand choice does not dramatically shift the shell to core BE or core CE values for stable NCs, corroborated by experiments that show virtually identical NC structures for a variety of substituted ligands (see Ref 39). Additionally, we note that for these NCs none of the methyl-substituted experimental NCs displayed significant rearrangement or deviation from the parity of Figure 2 apart from the $[\text{Ag}_{25}\text{SR}_{18}]^-$ NC. We have analyzed in detail the ligand effects in the $[\text{Ag}_{25}\text{SR}_{18}]^-$ NC in the Supplementary Information file and in our response 1 to reviewer 2.

Actions: To clarify ligand effects we have added Figure S6 and the following discussion focusing on only ligand-substitution effects:

Figure S6: Parity plot between core CE (kcal/mol) and the shell to core BE as suggested by our developed structure-dependent stability theory. Most of the values are identical to Figure 2. Additional points include: the $Ag_{25}SR_{18}^-$, with R=CH₃ (a) optimized and (b) experimental structures, respectively, (c) the optimized $Au_{18}SR_{14}$ with R=C₆H₁₁, and (d) the $[Au_{25}SR_{18}]^-$, with R=PhenylEthane structures. The silver arrow from (b) to (a) indicates the shift of the core CE and shell to core BE during optimization of the $[Ag_{25}SR_{18}]^-$ R=CH₃ NC. In the $[Ag_{25}SR_{18}]^-$ R=CH₃ NC image, the red ball/stick represent the experimental structure, whereas, the yellow sticks, the optimized structure.

Comments S6:

For the $[Ag_{25}SR_{18}]^-$ NC (S6 (a) and (b)), geometric reconstruction during optimization was noticeable (using as initial state the experimental structure and substituting the R-groups with methyls) and it is due to the lack of hydrogen-bonding in R-groups when R=CH₃. The surface reconstruction of the $[Ag_{25}SR_{18}]^-$ NC with methyls was also evident in the energetics of the NC after optimization when this was the only experimental structure that did not show the BE-CE energy balance. Since we noticed this reconstruction (and CE-BE imbalance), we considered the

full ligands and we optimized the experimental $[\text{Ag}_{25}(\text{SPhMe}_2)_{18}]^-$ NC. Only in this case, we noticed that optimizing the NC accounting for the full ligands results to a perfect CE-BE energy balance. It should be noticed that none of the experimental Au structures showed any similar reconstruction upon methyl substitution and optimization. To further verify that the methyl R-group substitution does not alter the stability of other Au NCs, we optimized the $\text{Au}_{18}\text{SR}_{14}$ and $[\text{Au}_{25}\text{SR}_{18}]^-$ NCs with their full ligands, finding variations of only 0.3 kcal/mol in the core to shell BE while the core CE remained identical to the $\text{R}=\text{CH}_3$ structure for each (S6 (c) and (d)).

We have also added a short discussion about the PhenylEthane $[\text{Au}_{25}\text{SR}_{18}]^-$ NC to the manuscript:

“For NC structures investigated in this work interactions at the interface between their core and shell regions should be to a large degree unaffected by the ligand selection⁴³ (see Figure S6 and Comments S6 where $\text{Au}_{18}\text{SR}_{14}$ and $[\text{Au}_{25}\text{SR}_{18}]^-$ optimization with full ligands resulted to minor energy shifts and for detailed analysis of the $[\text{Ag}_{25}\text{SR}_{18}]^-$ case).”

For these reasons I don't recommend publication of the present manuscript in Nature Communications.

PS Note that the literature citation is also incomplete. For example, electronic and charge analysis has been first performed in Ref. DOI: 10.1021/nn800268w (later work by Zeng's group are quoted).

Action: We have added this to the manuscript as reference #21

Ref. DOI: 10.1021/ja101083v is quoted to draw that the metal-metal interactions in the cores and shells of the nanoclusters are significantly stronger than the ligand-ligand dispersion interactions, which is true but hides the fact that residual effects are crucial to determine the differential stability of these species, outside trends with scaling size.

Response: We thank the reviewer for acknowledging that the metal-metal interactions are significantly stronger than the ligand-ligand interactions. Please also see our detailed response to the comment on the ligand effects.

Other stability analysis such as DOI: 10.1039/C2NR30501A (with the correlation between Au-S and shell to core binding strength)

Response: We thank the reviewer for making us aware of this interesting work.

Action: We have added this to the manuscript as reference. #23

and DOI: 10.1021/ja507738e (with energy decomposition analysis) are not mentioned whereas this work uses techniques there previously proposed.

Response: While the latter work is interesting, we have not used any energy decomposition analysis since we focus on the stability of different regions of the NCs (e.g. cores and core-shell interface) and not on type of energy contributions (e.g. electrostatic, dispersion etc.).

Action: We have added this article to the manuscript as reference #42
Further revisions not requested by reviewers:
We have updated the funding source in the acknowledgement section.

REVIEWERS' COMMENTS:

Reviewer #4 (Remarks to the Author):

The authors have revised and improved the manuscript, in the discussion and adding further data.

After careful reading the revised version and the rebuttal, I still believe that the good balance of Au-Au and Au-S bonding the authors find in this size range is interesting but coincidental, while the authors' arguments do not demonstrate that it is decisive.

Nevertheless, this observation may be useful, and I think the manuscript can now be accepted for publication on Nature Communications.